# Momentum Further Constrains Sharpness at the Edge of Stochastic Stability

**Arseniy Andreyev** [* 1]   **Advikar Ananthkumar** [* 2]   **Marc Walden** [* 2]   **Tomaso Poggio** [3]   **Pierfrancesco Beneventano** [3]

## Abstract

Recent work suggests that (stochastic) gradient descent self-organizes near an instability boundary, shaping both optimization and the solutions found. Momentum and mini-batch gradients are widely used in practical deep learning optimization, but it remains unclear whether they operate in a comparable regime of instability. We demonstrate that SGD with momentum exhibits an Edge of Stochastic Stability (EoSS)-like regime with *batch-size–dependent behavior* that cannot be explained by a single momentum-adjusted stability threshold. Batch Sharpness (the expected directional mini-batch curvature) stabilizes in two distinct regimes: at small batch sizes it converges to a lower plateau $2(1-\beta)/\eta$, reflecting amplification of stochastic fluctuations by momentum and favoring flatter regions than vanilla SGD; at large batch sizes it converges to a higher plateau $2(1+\beta)/\eta$, where momentum recovers its classical stabilizing effect and favors sharper regions consistent with full-batch dynamics. We further show that this aligns with linear stability thresholds and discuss the implications for hyperparameter tuning and coupling.

## 1. Introduction

**Optimization at the edge of stability.**   A growing body of evidence suggests that modern deep-network training with constant (or piecewise-constant) step size operates in a regime of *controlled instability*. In full-batch training, the top Hessian eigenvalue $\lambda_{\max}$ often sharpens until it hovers near a deterministic stability boundary (the *Edge of Stability*, EoS) (Xing et al., 2018; Jastrzębski et al., 2019; 2020; Cohen et al., 2021; 2025). In mini-batch training,

the full-batch sharpness $\lambda_{\max}$ can fail to diagnose stability; instead, Andreyev & Beneventano (2024) propose the *Edge of Stochastic Stability* (EoSS) and identify a directional mini-batch curvature statistic (Batch Sharpness, Definition 3.1) that saturates at $2/\eta$ for vanilla SGD.

**Where does momentum live relative to the stochastic edge?**   Momentum (Polyak heavy-ball, SGDM) and Nesterov acceleration (SGDN) are standard in deep learning and often essential for fast, stable training. Yet the "edge" picture is incomplete for momentum methods with mini-batch gradients: even in deterministic quadratics, momentum and Nesterov have different stability regions, and in the stochastic regime the relevant instability certificate is not obvious. For full-batch GD with momentum, Cohen et al. (2021) established cleanly that the stability threshold increases monotonically in $\beta$, namely as $\frac{2+2\beta}{\eta}$. We observe a consistent but surprising phenomenon in the mini-batch case.

*In small-batch settings, this trend reverses:*
*SGDM stabilization decreases monotonically in $\beta$.*

Moreover, SGD with momentum hovers in regions of extremely small curvature, where common "effective learning rate" heuristics are insufficient to predict which curvature levels or solutions training will select; see Figure 1. Following these observations, this paper asks:

*Does SGD with momentum or Nesterov acceleration self-organize at an instability boundary? If so, what boundary is actually being saturated?*

**Main empirical finding: momentum splits EoSS into two batch regimes.**   Across architectures and hyperparameters (Appendix J), the Batch Sharpness statistic *progressively sharpens* and then plateaus, but the plateau level depends sharply on batch size:

$$\mathrm{BS}_{\mathrm{plateau}} \approx \frac{2(1-\beta)}{\eta} \tag{1}$$

in the small-batch (noise-dominated) regime and

$$\mathrm{BS}_{\mathrm{plateau}} \approx \begin{cases} \frac{2(1+\beta)}{\eta} & \text{(SGDM)}, \\ \frac{2(1+\beta)}{\eta(1+2\beta)} & \text{(SGDN)} \end{cases} \tag{2}$$

---

*Equal contribution  [1]Princeton University, Princeton, NJ, USA. [2]Harvard University, Cambridge, MA, US [3]Massachusetts Institute of Technology, Cambridge, MA, US.. Correspondence to: Arseniy Andreyev <andreyev@princeton.edu>, Pierfrancesco Beneventano <pierb@mit.edu>.

*Proceedings of the 43rd International Conference on Machine Learning*, Seoul, South Korea. PMLR 306, 2026. Copyright 2026 by the author(s).

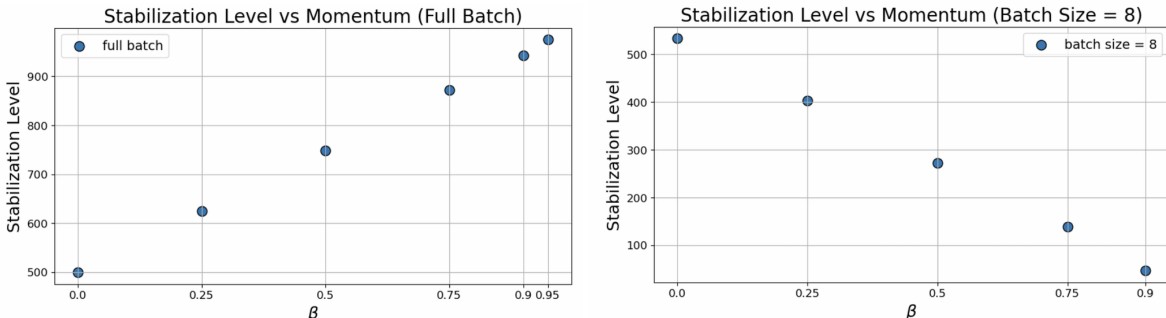

*Figure 1.* $\lambda_{\max}$ **under full-batch GD with momentum (left) and mini-batch SGD with momentum (right).** MLP on an 8k subset of CIFAR-10 for fixed step size $\eta = .004$ and varying $\beta$. The stabilization level of Batch Sharpness (Definition 3.1) inverts its monotonicity in $\beta$.

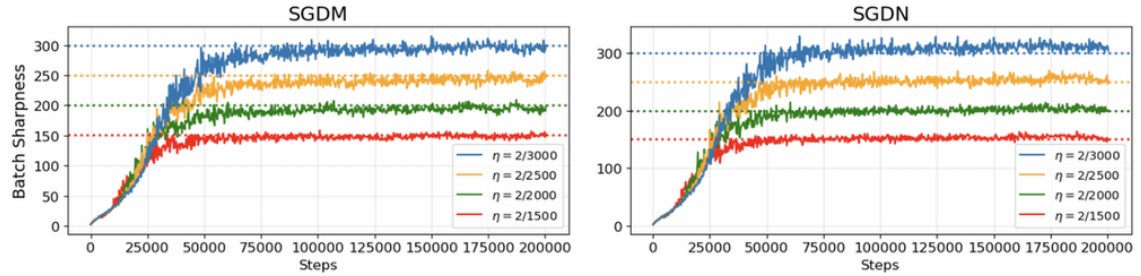

*Figure 2.* **EoSS phenomenon using SGDM (left) and SGDN (right).** MLPs on an 8k subset of CIFAR-10 under different step sizes $\eta$ and with $\beta = 0.9$. Batch Sharpness stabilizes around the $2(1 - \beta)/\eta = 1/(5\eta)$ threshold, shown by the dotted lines.

in the large-batch (deterministic) regime. [1] The small-batch plateau is *strictly lower* than the vanilla-SGD threshold $2/\eta$ and therefore reveals a qualitative "flip": with small batches, momentum enforces *stricter* curvature constraints and biases training toward flatter regions.

**Interventions certify an instability-adjacent regime.** To distinguish "mere plateaus" from genuine stability constraints, we use checkpoint interventions (Andreyev & Beneventano, 2024): small destabilizing changes to hyperparameters (e.g. $\eta \uparrow$, $b \downarrow$, or $\beta \uparrow$ in the small-batch regime) trigger catapult dynamics, followed by re-stabilization near the new plateau. This provides operational evidence that SGDM (and analogously SGDN) trains near an active stochastic stability boundary.

**Contributions.** More precisely, we establish:

- **Batch-size–dependent EoSS under momentum.** We show that SGDM and SGDN exhibit EoSS-like self-

[1]Here and below, statements that momentum permits higher large-batch curvature than vanilla SGD apply literally to SGDM; for SGDN, the deterministic plateau still exceeds the small-batch value $2(1 - \beta)/\eta$, but remains below the vanilla-SGD threshold $2/\eta$. This distinction does not affect the qualitative regime change: small batches tighten the operative curvature constraint, whereas large batches recover the optimizer-specific deterministic threshold.

organization, but the saturated curvature level depends fundamentally on batch size and cannot be explained by a single "momentum-corrected" threshold. Empirically, we identify a noise-dominated small-batch plateau at $2(1 - \beta)/\eta$ and a large-batch plateau approaching the deterministic momentum stability thresholds identified by Cohen et al. (2021) (with different large-batch limits for SGDM and SGDN), together with a broad intermediate regime interpolating between these two limits (Section 3.1).

- **Qualitative flip: small-batch momentum biases toward flatter regions.** The small-batch plateau at $2(1 - \beta)/\eta$ implies that, in sufficiently small batches, momentum does not relax the operative curvature constraint relative to vanilla SGD; it tightens it, biasing training toward flatter regions. This contrasts with the large-batch regime, where momentum recovers the classical stabilizing picture and permits sharper curvature (Section 3.1).

- **Intervention evidence for an EoSS-like regime of instability.** We show that destabilizing interventions produce catapults precisely when they push Batch Sharpness above its operating plateau, while stabilizing interventions reopen progressive sharpening. These intervention responses provide operational evidence that SGDM and SGDN train in an EoSS-like regime of instability, with Batch Sharpness as the operative diagnostic, rather

than merely exhibiting descriptive curvature plateaus (Section 3.3).

- **Mechanism via mean-square stability.** We provide a linear mean-square stability analysis showing that in the noise-dominated regime the instability threshold is governed by the effective step size $\eta_{\mathrm{eff}} = \eta/(1 - \beta)$. This explains the small-batch threshold $2(1-\beta)/\eta$ and is consistent with the observed interpolation across batch sizes. Importantly, this reduction concerns stability rather than full trajectory equivalence.[2] (Section 4).
- **Preliminary extension to Adam.** We provide evidence that the same two-regime role of momentum appears in Adam when viewed through preconditioned sharpness (Section 5.1).

## 2. Preliminaries and Related Work

### 2.1. Notation and Optimizers

We analyze mini-batch SGD with Polyak momentum or Nesterov acceleration on relatively simple vision classification tasks trained with MSE (see Section 6 for limitations). Precisely, let $\theta_t \in \mathbb{R}^d$ denote the model parameters at iteration $t$, let $\mathcal{D} = \{x_i\}_{i=1}^n$ be the dataset, and $\ell(\theta; x_i)$ be the loss on a sample $(x_i)$. We define the empirical risk

$$L(\theta) = \frac{1}{n} \sum_{i=1}^n \ell(\theta; x_i). \tag{3}$$

At each iteration, a mini-batch $\mathcal{B}_t$ of size $b$ is sampled uniformly at random, and the stochastic gradient is

$$g_t = \frac{1}{b} \sum_{i \in \mathcal{B}_t} \nabla_\theta \ell(\theta_t; x_i). \tag{4}$$

We use the heavy-ball (HB) momentum formulation standard in deep learning libraries, rather than an EMA-style update. The two algorithms considered are:

- SGD with Polyak Momentum (HB or SGDM):

$$\begin{aligned} v_{t+1} &= \beta v_t + g_t, \\ \theta_{t+1} &= \theta_t - \eta v_{t+1}, \end{aligned} \tag{5}$$

  with momentum $\beta \in [0, 1)$, learning rate $\eta > 0$, and $v_0 = 0$.
- SGD with Nesterov Acceleration (NAG or SGDN):

$$\begin{aligned} v_{t+1} &= \beta v_t + g_t \left(\theta_t - \beta \eta v_t\right), \\ \theta_{t+1} &= \theta_t - \eta v_{t+1}. \end{aligned} \tag{6}$$

Let $L_B(\theta) = \frac{1}{|B|} \sum_{i \in B} \ell(\theta; x_i)$ be the mini-batch loss for a batch $B \subseteq \mathcal{D}$ of size $b$ drawn from the mini-batch sampling distribution $\mathcal{P}_b$. Define the mini-batch gradient $g_B(\theta) = \nabla L_B(\theta)$ and mini-batch Hessian $H_B(\theta) = \nabla^2 L_B(\theta)$.

---

[2]Empirically, even when $\eta_{\mathrm{eff}}$ is matched so that curvature statistics stabilize similarly, SGD and SGDM remain separated in parameter/function space; see Appendix G.

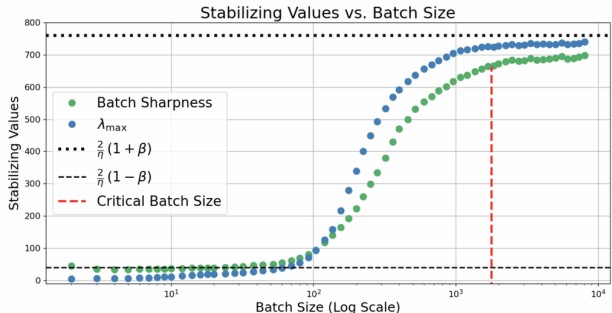

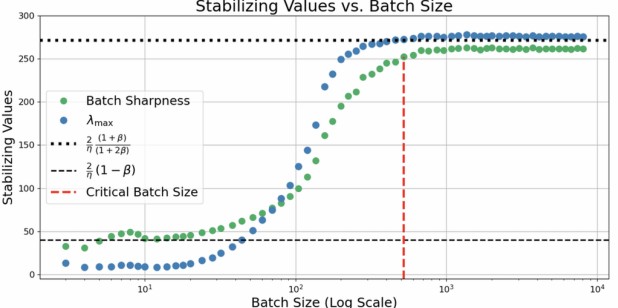

*Figure 3.* Stabilization levels of Batch Sharpness and $\lambda_{\max}$ across varying batch sizes for an MLP trained with SGDM (top) and SGDN (bottom) at $\eta = 0.005$ and $\beta = 0.9$. The critical batch size, defined heuristically as the threshold at which training dynamics enter the large-batch regime, is marked for each optimizer. Notably, SGDN reaches this regime at a batch size almost an order of magnitude smaller than SGDM.

### 2.2. The Value of Momentum

**The added value of momentum.** Polyak heavy-ball momentum and Nesterov acceleration are ubiquitous in modern deep learning—often as explicit buffers (SGDM/SGDN) or implicitly inside adaptive methods—and are frequently key to fast and stable training in practice (Krizhevsky et al., 2012; Sutskever et al., 2013; Gitman et al., 2019; Fu et al., 2023). A large body of work has proposed complementary explanations for why momentum helps (see Appendix B for further related work): *(i) stability enlargement / effective step-size rescaling*, where momentum can enlarge the usable learning-rate range, and in some regimes SGDM can be related to SGD after matching an effective step size (Fu et al., 2023; Wang et al., 2024; Gitman et al., 2019); *(ii) temporal filtering and noise-shaping*: the momentum buffer is an exponential moving average of stochastic gradients, motivating stationary-distribution, SDE, and modified-equation analyses (Mandt et al., 2017; Li et al., 2019; Gitman et al., 2019); *(iii) inertial/underdamped geometry*, where momentum is viewed as a discretization of a second-order flow and can therefore produce different transient dynamics than vanilla gradient descent (Su et al., 2014; Wibisono et al., 2016; Shi et al., 2022; Wilson et al., 2021); *(iv) solution selection and generalization*, where momentum can preserve or change implicit bias depending on regime and can affect generalization through implicit-regularization or stability

mechanisms (Wang et al., 2022; Jelassi & Li, 2022; Ghosh et al., 2023; Ramezani-Kebrya et al., 2024; Lyu et al., 2025). These perspectives motivate a wider central question:

*What are the effects of momentum and acceleration on the training dynamics?*

**What is known (and what remains unclear).** Recent work has also clarified that the effect of momentum is strongly regime dependent: *(1) small learning-rate regimes* can make momentum nearly redundant after matching effective learning rates, with SGD and SGDM often following closely tracking trajectories, or at least closely matching trajectory statistics, and showing limited additional gains (Fu et al., 2023; Wang et al., 2024); *(2) deterministic large learning-rate regimes* highlight momentum's stabilizing role, where it can substantially enlarge the range of usable learning rates and becomes most helpful near (or beyond) an instability boundary (Cohen et al., 2021); *(3) stochastic regimes* admit diffusion/modified-equation limits in which momentum changes how noise is accumulated over time and thus can influence exploration, stationary behavior, and sometimes implicit bias/generalization (Liu et al., 2021; Li et al., 2019; Jelassi & Li, 2022; Ramezani-Kebrya et al., 2024). At the same time, much of the theoretical momentum literature either (a) analyzes convergence/implicit bias under assumptions consistent with a *stable* descent-type regime, or (b) studies continuous-time limits that abstract away the batch-dependent curvature actually seen by mini-batch methods.

**EoS vs. mini-batch: why the full-batch $\lambda_{\max}$ signal breaks.** In full-batch GD, training typically self-organizes near the quadratic stability boundary $\lambda_{\max}(\nabla^2 L(\theta_t)) \approx 2/\eta$ and enters the oscillatory "central-flow" regime (Xing et al., 2018; Jastrzębski et al., 2019; 2020; Cohen et al., 2021; 2025). For deterministic heavy-ball momentum ($\beta$), the analogous linear boundary on quadratics is $\lambda_{\max} \approx 2(1 + \beta)/\eta$. In mini-batch training, however, this full-batch $\lambda_{\max}$ diagnostic does not carry over cleanly: $\lambda_{\max}(\nabla^2 L(\theta_t))$ can plateau far below $2/\eta$ (and may not stabilize), while loss oscillations are ubiquitous and by themselves do not diagnose instability (Cohen et al., 2021; Andreyev & Beneventano, 2024).

## 3. SGDM and SGDN Typically Operate at the Edge of Stochastic Stability

**The mechanism: instability criteria + perturbations.** Following Andreyev & Beneventano (2024), we treat an "edge" as *saturation of a computable one-sided instability certificate* for the local (quadratic) dynamics, and we *test* for this regime via checkpoint perturbations: restart from a checkpoint $\theta_t$ and apply a small destabilizing change (e.g. $\eta \uparrow$, $b \downarrow$, or $\beta \uparrow$). If training is near the active stability

boundary, the perturbed run exhibits a *catapult* (transient excursion + loss spike), whereas stabilizing perturbations do not. Note that this mechanism complements that of Cohen et al. (2021), who perturb the step size in the opposite direction to witness recovered stability and progressive sharpening. We use both mechanisms to establish that both SGDM and SGDN train at the Edge of Stochastic Stability.

We further test whether instability criteria (Definition B.1) known to govern optimizers without momentum or acceleration saturate here as well, and if so, at which level. On top of $\lambda_{\max}$, we track

**Definition 3.1** (*Batch Sharpness*). Assume the batches are drawn from the mini-batch sampling distribution $\mathcal{P}_b$. The *Batch Sharpness* at $\theta$ is

$$\text{BS}(\theta) := \mathbb{E}_{B \sim \mathcal{P}_b}\left[ \frac{g_B(\theta)^\top H_B(\theta)\, g_B(\theta)}{\|g_B(\theta)\|_2^2} \right]. \quad (7)$$

Andreyev & Beneventano (2024) showed that $\text{BS} > (2 + \varepsilon)/\eta$ is a sharp sufficient instability certificate for SGD, making BS the natural mini-batch analogue of the EoS curvature threshold. Further discussion appears in Appendix B.

Section 2 introduced the instability-centric viewpoint and the empirical diagnostics that we use throughout (including *Batch Sharpness*). Here we focus on a narrower question: *how do these dynamics change once momentum is introduced?* Concretely, we study SGD with Polyak (heavy-ball) momentum (SGDM) and with Nesterov acceleration (SGDN), as defined in Section 2.1.

The stability thresholds for *full-batch* gradient descent with momentum have been characterized in the classical optimization literature. For Heavy-Ball momentum, the stability boundary is $\lambda_{\max} < 2(1 + \beta)/\eta$ (Polyak, 1964), while for Nesterov acceleration it is $\lambda_{\max} < \frac{2(1+\beta)}{\eta(1+2\beta)}$ (Nesterov, 1988). Cohen et al. (2021) demonstrated that when using full-batch GD with momentum, $\lambda_{\max}$ indeed reaches and remains near these thresholds during training. However, it remains unclear whether an analogous regime of instability governs training when momentum is combined with mini-batch *stochastic* gradients. In this section, we establish that SGD with momentum (SGDM) and Nesterov acceleration (SGDN) train at the Edge of Stochastic Stability, and we characterize how this regime depends fundamentally on batch size.

### 3.1. Batch-Size–Dependent Curvature Plateau Under Momentum

We begin by examining within-run dynamics of Batch Sharpness for SGDM and SGDN. Across all settings we tested (architectures, activations, and hyperparameter sweeps; see Appendix J), Batch Sharpness and $\lambda_{\max}$ increase during the early stage of training and then plateau. The primary dif-

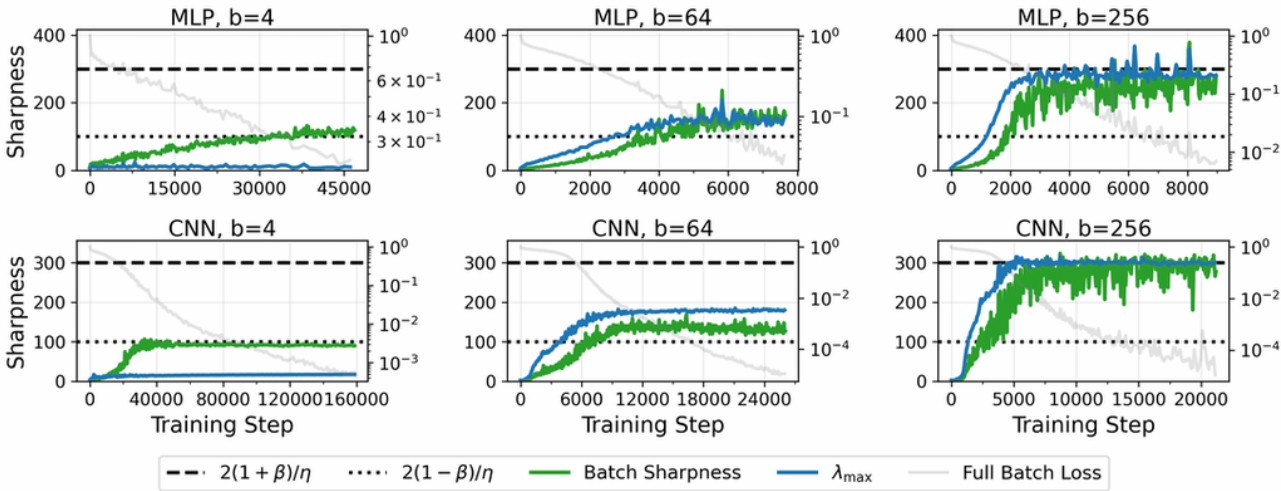

*Figure 4.* Dynamics of curvature statistics for SGDM with $\beta = 0.5$. Top row: MLP; bottom row: CNN. Columns correspond to batch sizes $b \in \{4, 64, 256\}$. Batch Sharpness and $\lambda_{\max}$ rise and then plateau, with larger batches yielding higher plateau levels. For Batch Sharpness, the left column is near the small-batch level $2(1 - \beta)/\eta$, the middle column lies in transition, and the right column approaches the large-batch level $2(1 + \beta)/\eta$.

ference from vanilla SGD is that the *plateau level* depends sharply on the batch size.

**Two regimes with a transition.** Empirically, two plateau regimes are consistently observed:

- **Small-batch regime.** For sufficiently small[3] $b$, the Batch Sharpness plateau is approximately

$$\text{BS}_{\text{plateau}} \approx \frac{2(1 - \beta)}{\eta}, \qquad (8)$$

for both SGDM and SGDN in our experiments (see Figure 4). Qualitatively, this means that the effect of momentum in small-batch SGD is opposite to its effect in full-batch GD: momentum now leads to *more restrictive* curvature levels.

- **Large-batch regime.** For large batch sizes we recover the full-batch behavior: *Batch Sharpness* and $\lambda_{\max}$ stabilize[4] at $\frac{2(1+\beta)}{\eta}$ for SGDM and at $\frac{2(1+\beta)}{\eta(1+2\beta)}$ for Nesterov's Accelerated Gradient (NAG), see Figure 3 and the right column of Figure 4. In this regime, momentum plays its classical role of allowing training in regions of higher curvature than its non-momentum counterparts.

---

[3]This is dataset-size-dependent, but for the 8k subset of CIFAR-10, "small" in this context means $b \lesssim 16$.

[4]Notice that Batch Sharpness sometimes stabilizes slightly below that threshold, consistent with (Andreyev & Beneventano, 2024); this can be explained by the fact that the full-batch gradient contains the self-stabilizing component of (Damian et al., 2023) in addition to the highest-eigenvector component.

Between these extremes, there is a broad transition region in which the plateau interpolates between the small-batch value and the large-batch value; see the middle column of Figure 4. Notably, the transition range we observe overlaps mini-batch sizes commonly used or studied in CIFAR-10 training (He et al., 2016; Zagoruyko & Komodakis, 2016; Masters & Luschi, 2018). The trend is monotone in $b$: larger batches yield systematically higher plateau levels (Figure 3). A second, more qualitative observation is that SGDM tends to transition later than SGDN: for fixed $(\eta, \beta)$, SGDN often reaches its large-batch (deterministic) plateau at smaller $b$, i.e., it tends to have a smaller critical batch size (Figure 3).

**Batch Sharpness as an indicator, not a certificate.** An important distinction from the vanilla SGD case must be emphasized. Andreyev & Beneventano (2024) established that for vanilla SGD, Batch Sharpness serves as an instability criterion—crossing $2/\eta$ is sufficient to guarantee divergence on the quadratic approximation. For SGDM/SGDN, we do not have an analogous theoretical result: Batch Sharpness does not necessarily govern the stability of the momentum dynamics in the same direct sense, as we further discuss in Section 4. Instead, Batch Sharpness here functions as an *empirical indicator* of a particular dynamical regime. Still, as discussed in Section 2, the precise way to establish whether the dynamics are in a regime of instability is through perturbation experiments, which we present in Section 3.3.

### 3.2. Consequences for $\lambda_{\max}$

Although we avoid interpreting Batch Sharpness as *the* stability quantity for momentum, it is still informative to track

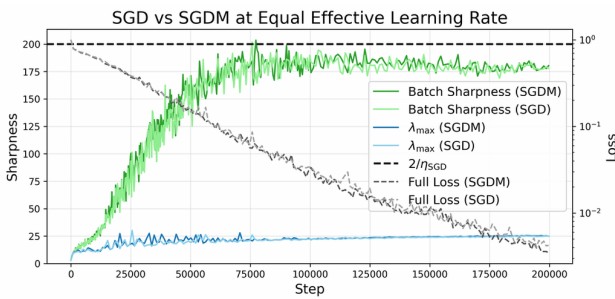

*Figure 5.* Within-run dynamics for an MLP with batch size $b = 4$. The SGDM run uses learning rate $\eta = 0.001$ with momentum $\beta = 0.9$, while the SGD run uses learning rate $\eta = 0.01$, chosen to match the effective step size.

how full-batch sharpness behaves alongside it. Empirically, as in the case of vanilla SGD, stabilization of Batch Sharpness induces a corresponding stabilization of the full-batch top eigenvalue $\lambda_{\max}$; see Figure 4 and Appendix J. Because Batch Sharpness stabilizes at $2(1 - \beta)/\eta$ in the *small-batch regime*, which is strictly lower than the vanilla SGD threshold of $2/\eta$, the full-batch eigenvalue $\lambda_{\max}$ is suppressed to even lower values than in vanilla SGD. This implies that *momentum with small batches biases training toward flatter regions* than either vanilla SGD or momentum with large batches.

**Matching stabilization levels.** In the small-batch regime, empirically, $SGDM(\eta, \beta, b)$ and vanilla $SGD(\frac{\eta}{1-\beta}, b)$ reach approximately the same $\lambda_{\max}$ stabilization level (Figure 5). This suggests that the stabilization level of Batch Sharpness is the primary determinant of where the full-batch eigenvalue $\lambda_{\max}$ settles, with Section 4 proposing a mechanism behind this behavior. Importantly, we observe in Appendix G that the two trajectories do not track each other; they merely share the same instability threshold.

**Two effects of increasing batch size.** As batch size increases, two concurrent effects raise the stabilization level of $\lambda_{\max}$. First, the stabilization threshold for Batch Sharpness itself increases from $2(1-\beta)/\eta$ toward $2(1+\beta)/\eta$. Second, the gap between Batch Sharpness and $\lambda_{\max}$ decreases and can even flip sign as batch size grows. Both effects push $\lambda_{\max}$ to stabilize at progressively higher values (Figures 3 and 4).

### 3.3. Showing Instability Through Interventions

The stabilization of Batch Sharpness at batch-size-dependent plateaus is suggestive of an EoSS-like regime, but does not by itself establish that SGDM and SGDN operate at an instability boundary. Following the discussion in Section 2, the definitive diagnostic for such a regime is the *intervention experiment*: if training self-organizes near an instability threshold, then small destabilizing perturbations

to hyperparameters should trigger characteristic *catapults*, i.e., abrupt loss spikes followed by restabilization. We now show that SGDM and SGDN exhibit precisely this behavior.

**Destabilizing perturbations trigger catapults.** We consider three classes of mid-training interventions that lower the effective stability threshold:

- **Increasing step size** $\eta \to \eta' > \eta$: this directly reduces the threshold from $2(1 - \beta)/\eta$ (small-batch) or $2(1 + \beta)/\eta$ (large-batch) to the corresponding value at $\eta'$.
- **Increasing momentum** $\beta \to \beta' > \beta$: in the small-batch regime, this tightens the constraint from $2(1 - \beta)/\eta$ to $2(1 - \beta')/\eta$; in the large-batch regime, the effect is reversed.
- **Decreasing batch size** $b \to b' < b$: this increases the value of Batch Sharpness, thus putting it above its stabilization threshold.

In each case, when the intervention causes the new threshold to fall below the current value of Batch Sharpness, we observe a catapult: a sharp spike in the training loss accompanied by a transient excursion in the curvature statistics (Figure 6). After the catapult, Batch Sharpness re-stabilizes around the new, lower threshold. This is the signature behavior of training at an instability boundary.

**Stabilizing perturbations restart progressive sharpening.** Conversely, although not crucial for establishing a regime of instability, interventions that raise the effective threshold—decreasing $\eta$, decreasing $\beta$, or increasing $b$—do not trigger catapults. Instead, these perturbations open a gap between the current Batch Sharpness and the new, higher threshold. This gap permits a renewed phase of progressive sharpening: Batch Sharpness gradually increases until it again approaches the updated threshold (Figure 11).

**Batch Sharpness, not $\lambda_{\max}$, governs the transition.** A key observation from these experiments is that the catapult/progressive-sharpening dynamics are predicted by Batch Sharpness, not by $\lambda_{\max}$. Specifically, when we change the *batch size* mid-training, $\lambda_{\max}$ does not change instantaneously—the full-batch loss landscape is unaffected by the choice of batch size. Yet we observe either a catapult or renewed progressive sharpening depending on whether Batch Sharpness crosses or falls below its stabilization level. This mirrors the findings of Andreyev & Beneventano (2024) for vanilla SGD and provides strong evidence that Batch Sharpness controls the stability of SGDM/SGDN dynamics.

A natural alternative would be to track mini-batch landscape sharpness, for example $\mathbb{E}_{B \sim \mathcal{P}_b}[\lambda_{\max}(H_B(\theta))]$. We do not use this quantity as the controlling statistic: Andreyev & Beneventano (2024) show that, even for vanilla SGD, it doesn't govern the dynamics. Since SGD is the special

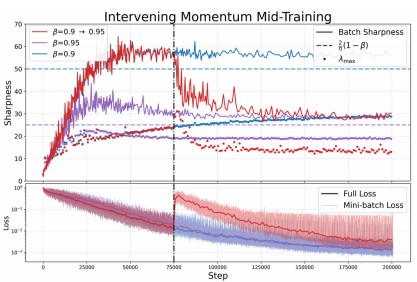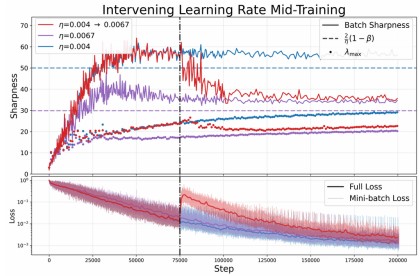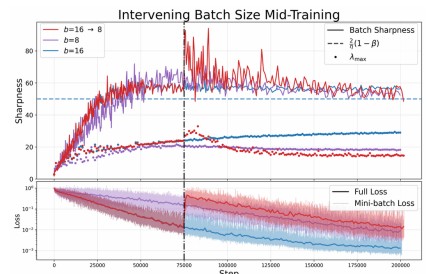

*Figure 6.* Within-run EoSS dynamics for an MLP under **destabilizing interventions** at step 75k with batch size $b = 16$, learning rate $\eta = 0.004$, and momentum $\beta = 0.9$. Left: destabilizing momentum intervention, increasing $\beta$ to 0.95. Middle: destabilizing learning rate intervention, increasing $\eta$ to 0.0067. Right: destabilizing batch size intervention, decreasing $b$ to 8. Top: Batch Sharpness and $\lambda_{\max}$. Bottom: Training loss.

case of SGDM with $\beta = 0$, mini-batch landscape sharpness cannot be the general stability-governing scalar for SGDM either.

**Instability without a complete theory.** We emphasize an important distinction from the vanilla SGD case. For SGD without momentum, Andreyev & Beneventano (2024) established that Batch Sharpness crossing $2/\eta$ is a valid instability criterion: on the quadratic approximation, exceeding this threshold guarantees divergence, see their Theorem 1. For SGDM and SGDN, we do not have an analogous theoretical result. Nevertheless, the empirical evidence confirms that SGDM and SGDN train at the edge of stochastic stability, potentially governed by Batch Sharpness:

- Batch Sharpness undergoes progressive sharpening and saturates at batch-size–dependent plateaus.
- Destabilizing interventions that push Batch Sharpness above its plateau trigger catapults.
- Stabilizing interventions that lower Batch Sharpness below its plateau restart progressive sharpening.
- These transitions occur precisely when Batch Sharpness crosses its stabilization level, independent of $\lambda_{\max}$.

The catapult-and-restabilization pattern is the operational signature of an instability boundary, even in the absence of a closed-form divergence theorem. Developing such a theory for momentum methods, including identifying the precise instability criterion and proving that it is saturated under progressive sharpening, remains an important open question.

## 4. The Instability Threshold

To understand the batch-size-dependent stability thresholds observed in Section 3, we perform a linear stability analysis following Wu et al. (2018); Ma & Ying (2021). We focus this formal development on SGDM, as the more commonly used momentum method. The key finding is that in the noise-dominated (small-batch) regime, the linearized mean-square stability of SGDM$(\eta, \beta)$ admits a leading-order reduction to

that of vanilla SGD with effective step size $\eta_{\text{eff}} := \eta/(1 - \beta)$.

**Setup: Linearization near a Minimizer.** Let $\theta^\star$ be an interpolating[5] minimizer with $\nabla \ell_i(\theta^\star) = 0$ for all $i$. Writing $x_t := \theta_t - \theta^\star$ and $H_i := \nabla^2 \ell_i(\theta^\star)$, the mini-batch Hessian is $\widehat{H}_t := \frac{1}{b} \sum_{j \in B_t} H_j$. Near $\theta^\star$, the SGDM update (5) linearizes to

$$v_t = \beta v_{t-1} + \widehat{H}_t x_{t-1}, \qquad x_t = x_{t-1} - \eta v_t. \quad (9)$$

We say that this linearized stochastic recursion is mean-square stable if the second moment of the state $(x_t, v_t)$ remains uniformly bounded over time.

### 4.1. Warm-Up: The One-Dimensional Case

We begin with a sketch of the one-dimensional case, with the details in Appendix C and D for SGDM and SGDN, respectively. When $d = 1$, or along any direction where the mini-batch Hessians $\{\widehat{H}_t\}$ commute, the system (9) reduces to a scalar second-order recursion with random curvature $h_t$ having mean $a := \mathbb{E}[h_t]$ and variance $\sigma_b^2 := \text{Var}(h_t)$. Analyzing mean-square stability via the induced recursion on $(\mathbb{E}[x_t^2], \mathbb{E}[x_t x_{t-1}], \mathbb{E}[x_{t-1}^2])$, the dominant eigenvalue of the mean-square operator admits the expansion

$$\lambda_\star(\eta) = 1 - \frac{\eta}{1-\beta} 2a + \frac{\eta^2}{(1-\beta)^2} \sigma_b^2 + O\left(\frac{\eta^2 a^2}{(1-\beta)^3}\right).$$

In the noise-dominated regime where $\sigma_b^2 \gg a^2/(1 - \beta)$, this recovers the same leading-order mean-square stability condition as vanilla SGD with step size $\eta_{\text{eff}} = \eta/(1 - \beta)$, as outlined in Wu et al. (2018).

**Interpolation between regimes.** In $d = 1$, we can derive the full stability boundary to showcase interpolation

---

[5]Relaxing the interpolation assumption to a general local minimum affects the steady-state variance of the iterates but leaves the fundamental stability criteria unchanged.

between the two limits:

$$\frac{1}{\eta_{\max}} = \underbrace{\frac{a}{2(1+\beta)}}_{\text{deterministic}} + \underbrace{\frac{\sigma_b^2}{2a(1-\beta)}}_{\text{stochastic}}. \tag{10}$$

When $\sigma_b^2 \to 0$ (large batch), this recovers the classical heavy-ball threshold $\eta_{\max} = 2(1+\beta)/a$. When $\sigma_b^2 \gg a^2$ (small batch), the stochastic term dominates, yielding the effective threshold $2(1-\beta)/\eta$ for the curvature. In between, this yields the interpolation observed empirically in Section 3.1.

### 4.2. Extension to Multiple Dimensions

The one-dimensional intuition extends to the general case. The mean-square dynamics of the augmented state $(x_t, v_t)$ are governed by a $4d^2 \times 4d^2$ linear operator whose spectrum splits into fast modes (contracting at rate $\beta$) and $d^2$ slow modes near unity. The following theorem, proved in Appendix E, characterizes the leading-order slow dynamics. Define the Kronecker moments and drift operator:

$$\bar{H} := \mathbb{E}[\widehat{H}_t], \quad G := \mathbb{E}[\widehat{H}_t \otimes \widehat{H}_t], \quad K := \bar{H} \otimes I_d + I_d \otimes \bar{H}$$

**Theorem 4.1.** *In the noise-dominated regime and for sufficiently small $\eta$, the leading-order mean-square stability condition for SGDM$(\eta, \beta)$ is:*

$$\rho\big(I_{d^2} - \eta_{\text{eff}} K + \eta_{\text{eff}}^2 G\big) < 1, \tag{11}$$

Notice that (11) coincides with the vanilla SGD stability condition of Ma & Ying (2021), Equation (31), evaluated at step size $\eta_{\text{eff}}$. We leave the details of the proof to Appendix E.

Crucially, theoretical links between SGDM stability and SGD with a modified step size already exist in the literature, see e.g. Yuan et al. (2016)[6] Our operator-centric approach recovers, to leading order, the vanilla-SGD stability criterion (rather than merely a sufficient condition); it also leaves open the possibility of explaining the empirically observed similarity of quantities like $\lambda_{\max}$ between SGDM and SGD through the explicit dependence on $\bar{H}$ in the constraint.

### 4.3. Connection to Batch Sharpness

Theorem 4.1 therefore suggests that small-batch SGDM should exhibit a stability threshold of $2/\eta_{\text{eff}} = 2(1-\beta)/\eta$, matching the empirical findings about *Batch Sharpness* stabilization in Section 3.1. In the noise-dominated small-batch regime, this link is indirect but theoretical: Theorem 4.1 reduces SGDM mean-square stability to vanilla SGD with step

---

[6]Although they assume smallness of the step size and prove strong approximation of SGDM and SGD trajectories, which does not hold for the "large" step sizes considered here; see Appendix G.

size $\eta_{\text{eff}}$, and Andreyev & Beneventano (2024) prove that for vanilla SGD Batch Sharpness is the governing instability criterion with threshold $2/\eta_{\text{eff}}$. Thus, in this regime, our theory supports the Batch Sharpness threshold through reduction to SGD. What is still missing is a direct momentum-specific bridge from the operator condition (11) to Batch Sharpness itself, especially outside the small-batch reduction.

## 5. Implications

We showed that SGDM and SGDN train neural networks at the Edge of Stochastic Stability and therefore inherit the implications previously established for GD, full-batch Adam, and SGD in the EoS literature (Cohen et al., 2021; 2022; 2025; Andreyev & Beneventano, 2024). The resulting implications include:

- **Descent-lemma proof templates fail at EoSS.** If training self-organizes near an instability boundary, uniform-smoothness arguments enforcing step-by-step monotone descent are typically not informative; this caveat extends to momentum methods once they exhibit EoSS-like plateaus.
- **What "large step size" means with momentum.** The relevant finite-$\eta$ comparator is *direction-aware* mini-batch curvature (Batch Sharpness), not the full-batch $\lambda_{\max}$. Under momentum, the operative plateau level is batch- and method-dependent (e.g., small-batch $BS_{\text{plateau}} \approx 2(1-\beta)/\eta$, while large-batch plateaus approach classical full-batch momentum thresholds). In particular, this reverses the full-batch intuition that momentum simply allows larger step sizes.
- **Stabilization becomes distributional.** Under progressive sharpening, "where the dynamics stabilizes" can depend on the *distribution* of mini-batch Hessians $\{H_B\}$ (and, with momentum, plausibly also on time-couplings induced by the velocity state), not only on the mean/full-batch Hessian.

On top of this, we established that momentum does not merely shift the edge of stability—it makes the edge itself batch-dependent, and in the stochastic regime it can enforce strictly flatter solutions than vanilla SGD.

**Momentum is not uniformly stabilizing.** In deterministic (large-batch) regimes, momentum recovers its classical stability enlargement and permits training near sharper curvature thresholds. In contrast, in noise-dominated (small-batch) regimes, momentum acts as an effective step-size amplifier, tightening the operative curvature constraint to $BS_{\text{plateau}} \approx 2(1-\beta)/\eta$ and biasing the search toward flatter regions.

**Instability viewpoint on hyperparameter coupling.** In the small-batch regime, $\eta_{\text{eff}} = \eta/(1-\beta)$ emerges as the leading-order stability parameter, so changing $\beta$ without a

compensating change in $\eta$ can move the dynamics across the stochastic edge. This provides a stability-centric rationale for coupling rules that approximately keep $\eta/(1 - \beta)$ fixed when varying momentum under noisy training.

**Toward adaptive instability control.** Interventions show that catapults occur when Batch Sharpness is pushed above its operating plateau, suggesting that Batch Sharpness tracking could serve as a practical instability monitor for tuning $\eta$, $\beta$, or $b$ to stay near—but not beyond—the stochastic edge.

### 5.1. The Role of Momentum in Other Optimizers

A natural extension is Adam, where momentum is coupled to adaptive preconditioning. The full-batch case is already known: Cohen et al. (2022) show that Adam stabilizes the preconditioned sharpness $\lambda_{\max}(P_t^{-1/2} H_t P_t^{-1/2})$ near $2/\eta \cdot (1 + \beta_1)/(1 - \beta_1)$. Thus full-batch Adam mirrors the full-batch SGDM momentum law, with preconditioned sharpness replacing raw sharpness and an extra $(1 - \beta_1)^{-1}$ from EMA-style momentum; see Appendix H.

For small-batch Adam, no Batch Sharpness analogue is known, so we use preconditioned $\lambda_{\max}$ as a proxy. The EMA normalization changes the form of the small-batch prediction: the stochastic $(1 - \beta_1)$ factor that lowers the SGDM threshold cancels the extra $(1 - \beta_1)^{-1}$ in Adam's EMA-style threshold. Thus, in the noise-dominated regime, the preconditioned-sharpness level should be approximately independent of $\beta_1$ rather than increasing with momentum. This is what our sweeps show (Figure 7): increasing $\beta_1$ raises the full-batch threshold, but the small-batch preconditioned $\lambda_{\max}$ trajectory stays nearly fixed and therefore sits progressively farther below the deterministic adaptive edge. In this sense, Adam exhibits the adaptive analogue of the SGDM reversal: momentum no longer stabilizes higher curvature, but acts as a noise accumulator that suppresses the attained curvature scale. Although Adam still lacks a stochastic instability criterion, this provides evidence that the instability-centric role of momentum is not specific to SGDM: the same two-regime structure appears in a practical adaptive optimizer.

## 6. Conclusion

We studied how momentum interacts with instability-limited mini-batch training. SGDM and SGDN exhibit an *Edge of Stochastic Stability* (EoSS)-like regime in which the saturated curvature is batch-dependent Batch Sharpness, rather than the full-batch Hessian spectrum.

The main effect is a batch-dependent reversal of the classical momentum intuition: large batches recover optimizer-specific deterministic thresholds, whereas small batches tighten the operative constraint to $2(1 - \beta)/\eta$, biasing training toward flatter regions. We confirm the presence of a

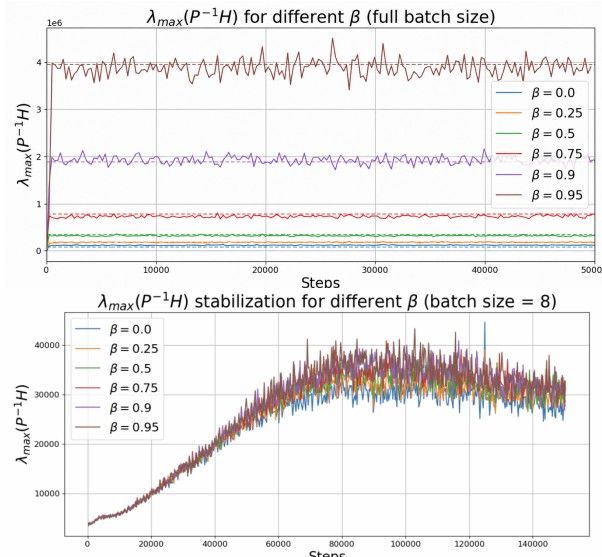

*Figure 7.* **Adam preconditioned sharpness across momentum.** Top: in full-batch Adam, $\lambda_{\max}(P^{-1}H)$ stabilizes near the EMA-momentum thresholds $\frac{2}{\eta} \frac{1+\beta_1}{1-\beta_1}$ (dashed). Bottom: at small batch size ($b = 8$), the trajectories are nearly independent of $\beta_1$, suggesting the presence of a similar noise-dominated effect of momentum in Adam.

regime of instability through destabilizing perturbation experiments. These results place momentum methods within the broader EoS/EoSS framework, and preliminary Adam experiments suggest a similar two-regime role in other optimizers. Code is available at this link.

### 6.1. Limitations and Future Work

Although our experiments cover CIFAR-10, SVHN, and SST, they remain limited to relatively small models (up to ResNet18 and a small ViT), reflecting the cost of distributional Hessian diagnostics at larger scale (Andreyev & Beneventano, 2024).

The main theoretical limitation is that Batch Sharpness is justified indirectly: in the noise-dominated small-batch regime, we reduce SGDM to SGD with effective step size $\eta_{\text{eff}}$ and invoke the SGD instability result of Andreyev & Beneventano (2024). A direct momentum-specific criterion, especially beyond this reduction and for SGDN, remains open.

Future work includes (i) finding a valid instability criterion for SGDM and SGDN, (ii) understanding the implications for learning and performance, (iii) identifying the sources of progressive sharpening and the self-stabilization mechanism, and (iv) extending this line of analysis to other optimizers such as Shampoo (Gupta et al., 2018) and Muon (Jordan et al., 2024).

## Impact Statement

This work clarifies the practical operating regime of momentum-based stochastic gradient methods in modern deep learning. Our findings have broader implications for optimizers and hyperparameter tuning, helping clarify the impact of momentum on optimization and generalization. We characterize deep learning optimization as shaped jointly by momentum and mini-batch noise, help reconcile disparate empirical observations, and provide a principled framework for understanding the dynamics of momentum in real-world training.

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

# Appendix

## A. Notation

| Symbol | Meaning |
| --- | --- |
| $\eta, \beta$ | Learning rate and momentum parameter. |
| $\eta_{\text{eff}}$ | Effective step size $\eta/(1-\beta)$ in the small-batch stability reduction. |
| $b, \mathcal{B}_t, B \sim \mathcal{P}_b$ | Mini-batch size, mini-batch at step $t$, and a generic batch drawn from the mini-batch sampling distribution. |
| $\theta_t \in \mathbb{R}^d, v_t$ | Model parameters and momentum/velocity buffer at optimization step $t$. |
| SGDM / HB | SGD with Polyak heavy-ball momentum. |
| SGDN / NAG | SGD with Nesterov acceleration. |
| $\mathcal{D} = \{x_i\}_{i=1}^n$ | Training set of $n$ samples. |
| $\ell(\theta; x_i)$ | Loss on sample $x_i$. |
| $L(\theta), L_B(\theta)$ | Empirical risk and mini-batch loss. |
| $g_t, g_B(\theta)$ | Sampled mini-batch gradient used at step $t$; for a generic batch $B$, $g_B(\theta) = \nabla L_B(\theta)$. |
| $H_B(\theta)$ | Mini-batch Hessian $\nabla^2 L_B(\theta)$. |
| $\lambda_{\max}$ | Top eigenvalue of $\nabla^2 L(\theta)$; the full-batch sharpness diagnostic. |
| $\text{BS}(\theta)$ | Batch Sharpness: $\mathbb{E}_{B \sim \mathcal{P}_b}\left[g_B(\theta)^\top H_B(\theta) g_B(\theta)/\|g_B(\theta)\|_2^2\right]$. |
| EoS, EoSS | Edge of Stability and Edge of Stochastic Stability. |
| $\theta^\star$ | Interpolating minimizer used as the expansion point in the local linear analysis. |
| $H_i, \widehat{H}_t$ | $H_i = \nabla_\theta^2 \ell(\theta^\star; x_i)$ and $\widehat{H}_t = 1/b \sum_{j \in \mathcal{B}_t} H_j$ in the linearized dynamics. |
| $h_t, a, \sigma_b^2$ | One-dimensional random curvature, its mean $\mathbb{E}[h_t]$, and its mini-batch variance $\text{Var}(h_t)$. |
| $\bar{H}, K, G$ | Stability-analysis operators: $\bar{H} = \mathbb{E}[\widehat{H}_t]$, $K = \bar{H} \otimes I_d + I_d \otimes \bar{H}$, and $G = \mathbb{E}[\widehat{H}_t \otimes \widehat{H}_t]$. |
| $\rho(\cdot)$ | Spectral radius. |

## B. Further Related Work

**The added value of momentum.** Polyak heavy-ball momentum and Nesterov-style acceleration are ubiquitous in modern deep learning and are often practically necessary for stable and fast training (Sutskever et al., 2013; Gitman et al., 2019; Fu et al., 2023). A large body of work has tried to isolate what momentum contributes beyond learning-rate tuning, and several complementary explanations have emerged. A first theme is *stability enlargement and step-size rescaling*: relative to vanilla SGD, adding momentum can enlarge the admissible learning-rate range by stabilizing the dynamics, and in certain regimes its net effect is well approximated by an effective learning rate $\eta_{\text{eff}} \approx \gamma/(1-\beta)$, so that SGD and SGDM can exhibit closely tracking trajectories (or statistics of trajectories) after matching $\eta_{\text{eff}}$ (Fu et al., 2023; Wang et al., 2024); related sufficient conditions can also be expressed in terms of global smoothness-type constants (Yuan et al., 2016). A second theme is *temporal filtering and noise shaping*: the momentum buffer is an exponential moving average of past stochastic gradients, $m_k = \sum_{j=0}^{k-1} \beta^{k-1-j} g_j$, so SGDM can be viewed as applying a linear time-invariant filter to gradient noise, motivating stationary-distribution, SDE-limit, and stochastic modified-equation analyses (Gitman et al., 2019; Li et al., 2017; 2019; Mandt et al., 2017). A third theme is *inertial (underdamped) dynamics and geometry*: momentum induces second-order behavior that can change transient exploration in narrow valleys and ill-conditioned directions relative to overdamped SGD; this viewpoint is often formalized via continuous-time limits and Lyapunov analyses (Su et al., 2014; Wibisono et al., 2016; Shi et al., 2022; Wilson et al., 2021; Liu et al., 2021). Beyond optimization speed, momentum can influence *solution selection and generalization* through implicit regularization and stability-based bounds (Wang et al., 2022; Jelassi & Li, 2022; Ghosh et al., 2023; Lyu et al., 2025; Ramezani-Kebrya et al., 2024), and it also interacts with *systems/implementation effects* (e.g., asynchrony), where staleness can induce implicit momentum and motivate "negative momentum" corrections (Mitliagkas et al., 2016).

**When does momentum help? A hyperparameter-dependent view.** Recent empirical and theoretical work suggests that the practical value of momentum is strongly regime-dependent. When the learning rate that yields good performance is already small—as is common in small/medium batch training and many fine-tuning setups—SGD and SGDM often have closely tracking trajectories after matching effective learning rates, and momentum can deliver only marginal gains in both optimization and generalization (Fu et al., 2023; Wang et al., 2024). In contrast, once one pushes toward larger step sizes, plain SGD often encounters an instability threshold first; in this high-step regime, momentum can matter substantially by enlarging the stability region and delaying or mitigating instability (Fu et al., 2023; Cohen et al., 2021).

In genuinely noisy regimes, momentum also changes the limiting stochastic dynamics (e.g., via underdamped diffusion limits and stochastic modified equations), affecting transient exploration and stationary behavior (Liu et al., 2021; Li et al., 2017; 2019; Mandt et al., 2017; Gitman et al., 2019). The picture for implicit bias and generalization is correspondingly mixed: in some separable linear settings momentum preserves the max-margin implicit bias (Wang et al., 2022), whereas in other regimes—including feature-learning settings, certain linear-network parameterizations, and multi-epoch stability analyses—momentum can provably or empirically change the selected solution and its test performance (Jelassi & Li, 2022; Ghosh et al., 2023; Lyu et al., 2025; Ramezani-Kebrya et al., 2024).

**Pointers into the momentum literature (deep-learning oriented).** On the empirical and algorithmic side, momentum schedules have long been motivated by their role in training large neural networks (Sutskever et al., 2013), with subsequent work proposing unifying parameterizations and guidelines (e.g., QHM/QHAdam) (Ma & Yarats, 2019) and characterizing the regimes in which momentum accelerates or becomes largely redundant (Fu et al., 2023; Wang et al., 2024). A complementary body of work uses dynamical-systems perspectives—ODE limits, variational formulations, "high-resolution" differential equations, and Lyapunov analyses—to explain how accelerated discretizations reshape stability and transient behavior (Su et al., 2014; Wibisono et al., 2016; Shi et al., 2022; Wilson et al., 2021). In stochastic regimes, diffusion approximations, stationary-law analyses, and stochastic modified-equation techniques provide a principled language for how momentum filters noise and induces underdamped dynamics (Liu et al., 2021; Li et al., 2017; 2019; Gitman et al., 2019; Mandt et al., 2017). Nonasymptotic convergence theory clarifies both limitations of vanilla momentum and circumstances where modified momentum schemes (often coupled with variance reduction) achieve provable gains (Kidambi et al., 2018; Liu et al., 2020; Cutkosky & Orabona, 2019; Paquette & Paquette, 2021). Finally, a growing literature studies momentum's effect on implicit bias and generalization in overparameterized learning (Jelassi & Li, 2022; Wang et al., 2022; Ghosh et al., 2023; Lyu et al., 2025; Ramezani-Kebrya et al., 2024), and systems work highlights how asynchrony can generate implicit momentum and how "negative momentum" can correct for staleness (Mitliagkas et al., 2016).

### B.1. Further Explanation: Edge of Stability: Deterministic Picture and Why Mini-Batch Is Different

**Progressive sharpening and EoS.** A sequence of works (Jastrzębski et al., 2019; 2020; Cohen et al., 2021) pinpoints a rapid early-time change in local curvature during training: along GD/SGD trajectories, the top Hessian eigenvalue $\lambda_{\max}$ typically exhibits a short initial dip, followed by a sustained increase. Jastrzębski et al. (2020) further report a sharp transition that ends this "progressive sharpening" phase. Subsequent evidence suggests that the timing of this transition is closely tied to optimizer stability (and can differ across algorithms even on the same objective) (Jastrzębski et al., 2019; 2020; Cohen et al., 2021; 2022). For full-batch methods, Cohen et al. (2021; 2022) relate the transition to the corresponding instability threshold; for instance, under GD, $\lambda_{\max}$ settles into oscillations around $2/\eta$, whereas for full-batch Adam (standard hyperparameters) the top eigenvalue of the *preconditioned* Hessian hovers around $38/\eta$. In MSE experiments, much of the remaining optimization appears to take place in this near-threshold regime, which in turn largely sets the $\lambda_{\max}$ of the final iterate (Cohen et al., 2021; 2022). Chen & Bruna (2023); Lee & Jang (2023) explain why $\lambda_{\max}$ can slightly exceed $2/\eta$ in practice: gradient nonlinearity introduces higher-order corrections that shift the effective boundary. The mechanism sustaining EoS-style dynamics has also been studied; Damian et al. (2023) argue that, under empirically supported alignment assumptions, third-order derivatives can provide a global stabilizing effect even when the local linearization is unstable.

**Full-batch EoS as a stability boundary (Cohen et al.).** A useful mental model is to view optimization as "surfing" the boundary of stability: the algorithm pushes toward higher curvature (which accelerates loss decrease) until it reaches the largest curvature that still permits sustained progress. Concretely, for full-batch gradient descent (GD) with constant step size $\eta$,

$$\theta_{t+1} = \theta_t - \eta \nabla L(\theta_t), \tag{12}$$

the local quadratic model $L(\theta) \approx L(\theta_t) + g^\top e + \frac{1}{2} e^\top H e$ predicts linear stability only if $\eta \, \lambda_{\max}(H) < 2$. Cohen et al. observed that, in deep networks, training often enters a regime where the full-batch sharpness $\lambda_{\max}(\nabla^2 L(\theta_t))$ increases ("progressive sharpening") until it hovers near the quadratic stability threshold, $\lambda_{\max}(\nabla^2 L(\theta_t)) \approx 2/\eta$, and training continues while remaining close to this boundary (the "Edge of Stability", EOS).

**Momentum near the deterministic edge.** For momentum methods (e.g. Polyak heavy-ball or Nesterov acceleration), classical linear analysis on quadratics predicts that the stability boundary shifts compared to GD, i.e. the maximal stable curvature increases by a $\beta$-dependent factor. Empirically, in deterministic or very-large-batch regimes this can manifest as an EoS-like plateau at a sharper level than $2/\eta$ (often summarized heuristically as a "$\sim (1 + \beta)$" increase in the plateau

sharpness), consistent with the intuition that momentum can tolerate and traverse higher-curvature directions when noise is small.

**Mini-batch training: why full-batch sharpness alone can be misleading.** Mini-batch optimizers do not evolve on a single fixed landscape: each step uses a sampled loss $L_B$ (with gradient $g_B$ and Hessian $H_B$), so stability is governed by the distribution of mini-batch geometries rather than only the full-batch Hessian. Practically, this means that the full-batch $\lambda_{\max}(\nabla^2 L(\theta_t))$ need not stabilize near $2/\eta$ in mini-batch regimes, even when training exhibits sharp transitions and intermittent "catapult"-like excursions. Moreover, stochastic training can exhibit persistent loss oscillations for reasons other than instability-limited dynamics, so the onset of oscillations is not, by itself, a reliable mini-batch analogue of the full-batch EoS signature. These differences motivate stability notions and diagnostics that directly target the stochastic dynamics.

**SGD stability analyses and a direction-aware mini-batch analogue of EoS.** Several papers (Wu et al., 2018; Ma & Ying, 2021; Granziol et al., 2022; Wu et al., 2022; Mulayoff & Michaeli, 2024) analyze constant-step-size SGD on quadratic objectives by studying *linear stability* of the parameter second moment, yielding sharp criteria of the form

$$\left\| \mathbb{E}_{B\sim\mathcal{P}_b}\big[(I - \eta H(L_B))^{\otimes 2}\big] \right\| \lessgtr 1.$$

These conditions can be interpreted as Lyapunov-style stability tests (e.g., for the squared distance to the minimizer) and clarify how sampling-induced randomness alters stability relative to full-batch GD. In deep networks, directly evaluating such $d^2$-dimensional operators is typically impractical, so a parallel literature explores more tractable diagnostics and proxies, including scalar curvature summaries such as trace-style Hessian/NTK quantities (Wu & Su, 2023; Agarwala & Pennington, 2024) and empirical characterizations of oscillatory regimes (Cohen et al., 2021; Xing et al., 2018; Lee & Jang, 2023; Ahn et al., 2022). A complementary and especially simple direction-aware statistic is the curvature *along the stochastic update direction*, captured by the *Batch Sharpness*:

$$\mathrm{BS}(\theta) := \mathbb{E}_{B\sim P_b}\left[\frac{g_B(\theta)^\top H_B(\theta)\, g_B(\theta)}{\|g_B(\theta)\|^2}\right], \qquad g_B(\theta) = \nabla L_B(\theta),\ H_B(\theta) = \nabla^2 L_B(\theta). \tag{13}$$

$\mathrm{BS}(\theta)$ is the expected Rayleigh quotient of the *mini-batch* Hessian along the *mini-batch* gradient direction; informally, it is the expected curvature "felt" along the stochastic update. Under a local quadratic approximation, for vanilla SGD the threshold $\mathrm{BS}(\theta) > 2/\eta$ (by any fixed margin) is sufficient to trigger a catapult-like excursion, suggesting a mini-batch edge-of-stochastic-stability condition

$$\textbf{EoSS:} \qquad \mathrm{BS}(\theta_t) \text{ progressively sharpens and then hovers near } 2/\eta. \tag{14}$$

For momentum methods, the exact closed-form instability criterion is more delicate; nevertheless, direction-aware curvature tracking (e.g., $\mathrm{BS}(\theta_t)$) together with a targeted intervention test (below) provides a practical way to diagnose whether training is genuinely constrained by an instability boundary.

**B.2. More on the Mechanism: Instability Criteria and an Intervention**

**From local descent models to instability criteria (what an "edge" means stochastically).** In stochastic optimization, insisting on a single global descent lemma is often too blunt; a common alternative is a local viewpoint: fix a checkpoint $\theta_t$ and a neighborhood $U_t$ where a local approximation (e.g., a quadratic model) is believed to be informative. Within such a neighborhood, one can formalize the notion of a stability boundary through an algorithm-dependent *instability criterion*.

**Definition B.1** (Instability criterion). Consider a training algorithm (a discrete-time dynamical system) $(\theta_t)_{t\geq 0}$ on a parameter space $\Theta \subseteq \mathbb{R}^d$ with fixed hyperparameters $h$ (e.g. learning rate, batch size). Let $U \subseteq \Theta$ be an open set (typically, a region where a local approximation of the loss is trusted), and let $f : U \to \mathbb{R}$ and $c \in \mathbb{R}$. We say that $f$ is a *valid instability criterion with threshold $c$* for the algorithm on $U$ if

$$f(\theta_0) > c \implies (\theta_t)_{t\geq 0} \text{ leaves every compact subset of } U.$$

Equivalently: for any compact $K \subset U$ containing $\theta_0$, there exists a finite time $T$ such that $\mathbb{P}(\theta_T \notin K \mid \theta_0) > 0$. We say that $f$ is *saturated* at $\theta$ if $f(\theta)$ is (approximately) equal to $c$; in practice, up to an $O(\eta \cdot \mathrm{poly}(\log(\eta)))$ tolerance.

Stochastic systems can admit multiple valid instability criteria (depending on which local model and which Lyapunov function one uses), so the "edge" is best understood operationally: training self-organizes so that *some* valid criterion saturates, $f(\theta_t; h) \approx c(h)$, under progressive sharpening. A plateau in a diagnostic plot can therefore be suggestive, but is not, by itself, conclusive evidence that training is truly instability-limited.

**A checkpoint perturbation test for "training at the edge".** An intervention-based mechanism can make "training at the edge" more directly falsifiable. Take a checkpoint $\theta_t$ from a baseline run with hyperparameters $h_0$, and restart training from $\theta_t$ under a *small destabilizing perturbation* of $h_0$ (e.g. $\eta \uparrow$ or $b \downarrow$, with other settings fixed). If the baseline dynamics is genuinely stability-limited at $\theta_t$, then this small perturbation typically triggers a rapid transient runaway (a "catapult"): a large excursion accompanied by a sharp loss spike, followed (in many cases) by re-stabilization and saturation at a new level consistent with the perturbed hyperparameters. Conversely, *stabilizing* perturbations (e.g. $\eta \downarrow$ or $b \uparrow$) should suppress catapults and can reopen a "gap" that allows renewed progressive sharpening. This intervention test is useful precisely because it distinguishes quantities that merely *appear* to plateau from quantities that are actually governing local instability: if a diagnostic seems to saturate but small destabilizing perturbations do not trigger catapults, then that diagnostic is unlikely to be a valid instability criterion for the relevant stochastic dynamics.

## C. Proof of the 1D Warm-Up Case for SGDM

This Appendix provides the proofs for the claims for the one-dimensional case in Section 4.1.

We consider the one-dimensional linearized SGDM:

$$x_t = (1 + \beta - \eta h_t)x_{t-1} - \beta x_{t-2}, \tag{15}$$

where $(h_t)_{t \geq 1}$ are i.i.d., independent of $(x_{t-1}, x_{t-2})$, with

$$a := \mathbb{E}[h_t], \qquad \sigma_b^2 := \mathrm{Var}(h_t).$$

(Under without-replacement mini-batching, $\sigma_b^2 = \frac{n-b}{b(n-1)}\sigma^2$ with $\sigma^2 := \frac{1}{n}\sum_{i=1}^n a_i^2 - a^2$.)

**Closed recursion for second moments.** Let $\alpha_t := 1 + \beta - \eta h_t$ and define the second-moment state

$$w_t := \begin{bmatrix} \mathbb{E}[x_t^2] \\ \mathbb{E}[x_t x_{t-1}] \\ \mathbb{E}[x_{t-1}^2] \end{bmatrix}.$$

Thus,

$$x_t^2 = \alpha_t^2 x_{t-1}^2 - 2\beta \alpha_t x_{t-1} x_{t-2} + \beta^2 x_{t-2}^2, \qquad x_t x_{t-1} = \alpha_t x_{t-1}^2 - \beta x_{t-1} x_{t-2}.$$

Combined with

$$\alpha_1 := \mathbb{E}[\alpha_t] = 1 + \beta - \eta a, \qquad \alpha_2 := \mathbb{E}[\alpha_t^2] = (1 + \beta - \eta a)^2 + \eta^2 \sigma_b^2 = \alpha_1^2 + \eta^2 \sigma_b^2,$$

we get

$$w_t = R(\eta)\, w_{t-1}, \qquad R(\eta) := \begin{bmatrix} \alpha_2 & -2\beta\alpha_1 & \beta^2 \\ \alpha_1 & -\beta & 0 \\ 1 & 0 & 0 \end{bmatrix}. \tag{16}$$

Mean-square linear stability is thus equivalent to $\rho(R(\eta)) < 1$

**Perturbative expansion.** Characteristic polynomial of $\rho(R(\eta))$:

$$p_\eta(\lambda) = \lambda^3 + \left(\beta - \alpha_2\right)\lambda^2 + \left(2\beta\alpha_1^2 - \beta\alpha_2 - \beta^2\right)\lambda - \beta^3. \tag{17}$$

At $\eta = 0$, $\alpha_1 = 1 + \beta$ and $\alpha_2 = (1 + \beta)^2$, and (17) factorizes as

$$p_0(\lambda) = (\lambda - 1)(\lambda - \beta)(\lambda - \beta^2),$$

so the only eigenvalue on the unit circle is $\lambda = 1$. By continuity, for small $\eta$ there is a unique eigenvalue $\lambda_\star(\eta)$ with $\lambda_\star(0) = 1$ governing the stability boundary.

To expand $\lambda_\star(\eta)$, set the ansatz

$$\lambda_\star(\eta) = 1 + c_1\eta + c_2\eta^2 + O(\eta^3),$$

and impose $p_\eta(\lambda_\star(\eta)) \equiv 0$. Expanding (17) in $\eta$ and matching coefficients yields

$$c_1 = -\frac{2a}{1 - \beta}, \qquad c_2 = \frac{\sigma_b^2}{(1 - \beta)^2} + \frac{1 - 3\beta}{(1 - \beta)^3}a^2.$$

Therefore,

$$\lambda_\star(\eta) = 1 - \frac{2a}{1 - \beta}\eta + \left(\frac{\sigma_b^2}{(1 - \beta)^2} + \frac{1 - 3\beta}{(1 - \beta)^3}a^2\right)\eta^2 + O(\eta^3). \tag{18}$$

In the noise-dominated regime where $\frac{a^2}{1-\beta} \ll \sigma_b^2$ (so the $a^2$-term is lower order relative to $\sigma_b^2$), this reduces to

$$\lambda_\star(\eta) = 1 - 2a\,\eta_{\mathrm{eff}} + \sigma_b^2\,\eta_{\mathrm{eff}}^2 + o(\eta^2), \qquad \eta_{\mathrm{eff}} := \frac{\eta}{1 - \beta},$$

which is the same leading-order stability condition as in the Wu et al. (2018) analysis, with stepsize $\eta_{\mathrm{eff}}$.

**Exact interpolation formula for $\eta_{\max}$.** The convenience of 1D case is that we compute the exact interpolation case. Since the eigenvalues are $\{1, \beta, \beta^2\}$ at $\eta = 0$ and $\beta, \beta^2 < 1$, the first loss of stability occurs when the dominant mode crosses at $\lambda = 1$. The Jury conditions for (17) reduce to the single active inequality $p_\eta(1) > 0$. Evaluating (17) at $\lambda = 1$ and simplifying gives

$$p_\eta(1) = \eta\Big(2a(1 + \beta)(1 - \beta) - \eta\big((1 - \beta)a^2 + (1 + \beta)\sigma_b^2\big)\Big).$$

Thus $p_\eta(1) > 0$ holds iff $0 < \eta < \eta_{\max}$, where

$$\eta_{\max} = \frac{2a(1 + \beta)(1 - \beta)}{(1 - \beta)a^2 + (1 + \beta)\sigma_b^2} = \frac{2a(1 + \beta)}{a^2 + \frac{1+\beta}{1-\beta}\sigma_b^2}. \tag{19}$$

For convenience, we can take the reciprocal

$$\frac{1}{\eta_{\max}} = \frac{a}{2(1 + \beta)} + \frac{\sigma_b^2}{2a(1 - \beta)}. \tag{20}$$

The deterministic limit $\sigma_b^2 \to 0$ yields $\eta_{\max} = 2(1 + \beta)/a$, while the noise-dominated limit $\sigma_b^2 \gg a^2$ yields

$$\eta_{\max} \approx \frac{2a(1 - \beta)}{\sigma_b^2} \quad \Longleftrightarrow \quad \eta_{\mathrm{eff},\max} := \frac{\eta_{\max}}{1 - \beta} \approx \frac{2a}{\sigma_b^2},$$

matching the 1D SGD noise-dominated threshold with the effective stepsize $\eta_{\mathrm{eff}} = \eta/(1 - \beta)$.

# D. Proof of the 1D Warm-Up Case for SGDN

This Appendix provides the proofs for the claims for the one-dimensional case of SGD with Nesterov momentum (SGDN) in the sense of (6). This is essentially repeating Appendix C, but for SGDN, showcasing the same stability threshold.

We consider the 1D quadratic linearization near a global minimizer $x^\star = 0$, for which the (mini-batch) gradient evaluated at any point $y \in \mathbb{R}$ takes the form $g_t(y) = h_t y$, where $(h_t)_{t \geq 0}$ are i.i.d., independent of the past iterates, with

$$a := \mathbb{E}[h_t], \qquad \sigma_b^2 := \mathrm{Var}(h_t).$$

(Under without-replacement mini-batching, $\sigma_b^2 = \frac{n-b}{b(n-1)}\sigma^2$ with $\sigma^2 := \frac{1}{n}\sum_{i=1}^n a_i^2 - a^2$.)

**Deriving the 2-Step Recursion**

Let $x_t := \theta_t - \theta^\star = \theta_t$. The NAG update is

$$v_{t+1} = \beta v_t + g_t(\theta_t - \beta\eta v_t) = \beta v_t + h_t(x_t - \beta\eta v_t), \qquad x_{t+1} = x_t - \eta v_{t+1}.$$

Hence

$$v_{t+1} = h_t x_t + \beta(1 - \eta h_t)v_t, \qquad x_{t+1} = x_t - \eta h_t x_t - \beta\eta(1 - \eta h_t)v_t = (1 - \eta h_t)\big(x_t - \beta\eta v_t\big).$$

Using $x_t = x_{t-1} - \eta v_t$, i.e. $v_t = (x_{t-1} - x_t)/\eta$, we obtain

$$x_t - \beta\eta v_t = x_t - \beta(x_{t-1} - x_t) = (1 + \beta)x_t - \beta x_{t-1}.$$

Therefore the 1D linearized NAG dynamics reduces to the random-coefficient 2-step recursion

$$\boxed{x_{t+1} = (1 - \eta h_t)\big((1 + \beta)x_t - \beta x_{t-1}\big).} \tag{21}$$

**Closed Recursion for Second Moments**

Let $r_t := 1 - \eta h_t$, and define the second-moment state

$$w_t := \begin{bmatrix} \mathbb{E}[x_t^2] \\ \mathbb{E}[x_t x_{t-1}] \\ \mathbb{E}[x_{t-1}^2] \end{bmatrix}.$$

From (21) we have

$$x_{t+1}^2 = r_t^2 \Big( (1 + \beta)^2 x_t^2 - 2\beta(1 + \beta)x_t x_{t-1} + \beta^2 x_{t-1}^2 \Big),$$

$$x_{t+1}x_t = r_t \Big( (1 + \beta)x_t^2 - \beta x_t x_{t-1} \Big).$$

Define the first two moments of $r_t$:

$$p := \mathbb{E}[r_t] = 1 - \eta a, \qquad q := \mathbb{E}[r_t^2] = \mathbb{E}\big[(1 - \eta h_t)^2\big] = (1 - \eta a)^2 + \eta^2 \sigma_b^2 = p^2 + \eta^2 \sigma_b^2.$$

We get

$$w_{t+1} = R_{\mathrm{NAG}}(\eta)\, w_t, \qquad R_{\mathrm{NAG}}(\eta) := \begin{bmatrix} (1+\beta)^2 q & -2\beta(1+\beta)q & \beta^2 q \\ (1+\beta)p & -\beta p & 0 \\ 1 & 0 & 0 \end{bmatrix}. \tag{22}$$

Mean-square linear stability is thus equivalent to $\rho(R_{\mathrm{NAG}}(\eta)) < 1$.

### Perturbative Expansion of the Dominant Eigenvalue

Let $p_\eta(\lambda) := \det(\lambda I - R_{\mathrm{NAG}}(\eta))$. A direct determinant calculation gives the characteristic polynomial

$$p_\eta(\lambda) = \lambda^3 + \Big(\beta p - (1+\beta)^2 q\Big)\lambda^2 + \beta q\Big((1+\beta)^2 p - \beta\Big)\lambda - \beta^3 pq. \tag{23}$$

At $\eta = 0$, $p = q = 1$ and (23) factorizes as

$$p_0(\lambda) = (\lambda - 1)(\lambda - \beta)(\lambda - \beta^2),$$

so the only eigenvalue on the unit circle is $\lambda = 1$. By continuity, for small $\eta$ there is a unique eigenvalue $\lambda_\star(\eta)$ with $\lambda_\star(0) = 1$ governing the stability boundary.

To expand $\lambda_\star(\eta)$, set the ansatz

$$\lambda_\star(\eta) = 1 + c_1 \eta + c_2 \eta^2 + O(\eta^3),$$

and impose $p_\eta(\lambda_\star(\eta)) \equiv 0$. Expanding (23) in $\eta$ and matching coefficients yields

$$c_1 = -\frac{2a}{1 - \beta}, \qquad c_2 = \frac{\sigma_b^2}{(1-\beta)^2} + \frac{1 - \beta - 2\beta^2}{(1-\beta)^3}\, a^2.$$

Therefore,

$$\lambda_\star(\eta) = 1 - \frac{2a}{1 - \beta}\, \eta + \left( \frac{\sigma_b^2}{(1-\beta)^2} + \frac{1 - \beta - 2\beta^2}{(1-\beta)^3}\, a^2 \right) \eta^2 + O(\eta^3). \tag{24}$$

In the noise-dominated regime where $\frac{a^2}{1-\beta} \ll \sigma_b^2$ (so the $a^2$-term is lower order relative to $\sigma_b^2$), (24) reduces to

$$\lambda_\star(\eta) = 1 - 2a\, \eta_{\mathrm{eff}} + \sigma_b^2\, \eta_{\mathrm{eff}}^2 + o(\eta^2), \qquad \eta_{\mathrm{eff}} := \frac{\eta}{1 - \beta},$$

which matches the leading-order stability condition of 1D SGD with stepsize $\eta_{\mathrm{eff}}$, just like in the case of SGDM.

# E. Proofs for the Multi-Dimensional Case

In this appendix, we provide the proof that the stability of $\text{SGDM}(\eta, \beta)$ reduces to that of $\text{SGD}(\eta_{\text{eff}})$ in the multi-dimensional case with non-commuting mini-batch Hessians.

*Proof of Theorem 4.1.* Throughout, write $H_t := \widehat{H}_t$ and assume $\{H_t\}_{t \geq 1}$ are i.i.d. with finite second moment and independent of $(x_{t-1}, v_{t-1})$. Define

$$\bar{H} := \mathbb{E}[H_t], \qquad K := \bar{H} \otimes I + I \otimes \bar{H}, \qquad G := \mathbb{E}[H_t \otimes H_t].$$

Consider the linearized SGDM recursion (9),

$$v_t = \beta v_{t-1} + H_t x_{t-1}, \qquad x_t = x_{t-1} - \eta v_t.$$

Introduce the augmented state $z_t := \begin{bmatrix} x_t \\ v_t \end{bmatrix} \in \mathbb{R}^{2d}$, so that

$$z_t = A_t z_{t-1}, \qquad A_t := \begin{bmatrix} I - \eta H_t & -\eta \beta I \\ H_t & \beta I \end{bmatrix}. \tag{25}$$

**Second-moment operator.** Let $\Sigma_t := \mathbb{E}[z_t z_t^\top]$ and $m_t := \text{vec}(\Sigma_t)$. Vectorizing, we get:

$$m_t = \mathcal{T}_{\text{HB}}(\eta, \beta) m_{t-1}, \quad \text{where} \quad \mathcal{T}_{\text{HB}}(\eta, \beta) := \mathbb{E}[A_t \otimes A_t]. \tag{26}$$

Mean-square stability is then equivalent to $\rho(\mathcal{T}_{\text{HB}}(\eta, \beta)) < 1$.

Partition $m_t$ into the four $(d^2)$-blocks

$$m_t^{xx} := \mathbb{E}[x_t \otimes x_t], \quad m_t^{xv} := \mathbb{E}[x_t \otimes v_t], \quad m_t^{vx} := \mathbb{E}[v_t \otimes x_t], \quad m_t^{vv} := \mathbb{E}[v_t \otimes v_t],$$

and set $u_t := m_t^{xx}$ and $y_t := \left(m_t^{xv}, m_t^{vx}, m_t^{vv}\right)^\top \in \mathbb{R}^{3d^2}$. Then (26) can be written in block form

$$\begin{bmatrix} u_t \\ y_t \end{bmatrix} = \begin{bmatrix} M_{11}(\eta) & M_{12}(\eta) \\ M_{21}(\eta) & M_{22}(\eta) \end{bmatrix} \begin{bmatrix} u_{t-1} \\ y_{t-1} \end{bmatrix}. \tag{27}$$

A direct expansion of $\mathcal{T}_{\text{HB}}(\eta, \beta) = \mathbb{E}[A_t \otimes A_t]$ yields

$$M_{11}(\eta) = \mathbb{E}[(I - \eta H_t) \otimes (I - \eta H_t)] = I_{d^2} - \eta K + \eta^2 G, \tag{28}$$

and $M_{12}(\eta) = O(\eta)$, $M_{21}(\eta) = O(1)$, while

$$M_{22}(0) = \begin{bmatrix} \beta I_{d^2} & 0 & 0 \\ 0 & \beta I_{d^2} & 0 \\ \beta(\bar{H} \otimes I) & \beta(I \otimes \bar{H}) & \beta^2 I_{d^2} \end{bmatrix}, \qquad \rho\left(M_{22}(0)\right) = |\beta| < 1. \tag{29}$$

Hence for $\eta$ sufficiently small, $\rho(M_{22}(\eta)) \leq |\beta| + O(\eta) < 1$.

**Schur-complement reduction.** Since $M_{22}(\eta)$ is strictly stable, the eigenvalues of the full operator in a neighborhood of 1 are governed by the Schur complement:

$$T_{\text{slow}}(\eta, \beta) := M_{11}(\eta) + M_{12}(\eta)\left(I_{3d^2} - M_{22}(\eta)\right)^{-1} M_{21}(\eta), \tag{30}$$

in the sense that

$$\rho\left(\mathcal{T}_{\text{HB}}(\eta, \beta)\right) < 1 \quad \Longleftrightarrow \quad \rho\left(T_{\text{slow}}(\eta, \beta)\right) < 1, \tag{31}$$

and the remaining spectrum of $\mathcal{T}_{\text{HB}}(\eta, \beta)$ stays uniformly bounded by $|\beta| + O(\eta)$.

**Small-$\eta$ expansion.** Expanding (30) around $\eta = 0$ and retaining terms up to $O(\eta^2)$ gives

$$T_{\text{slow}}(\eta, \beta) = I_{d^2} - \frac{\eta}{1 - \beta} K + \frac{\eta^2}{(1 - \beta)^2} G + R(\eta, \beta), \tag{32}$$

where $R(\eta, \beta)$ collects the (deterministic) curvature-squared contributions and higher-order terms; a crude norm bound of the correct scaling is

$$\|R(\eta, \beta)\| \lesssim \frac{\eta^2}{(1 - \beta)^3} \|\bar{H}\|^2. \tag{33}$$

Define $\eta_{\text{eff}} := \eta / (1 - \beta)$. Then (32) reads

$$T_{\text{slow}}(\eta, \beta) = I_{d^2} - \eta_{\text{eff}} K + \eta_{\text{eff}}^2 G + R(\eta, \beta). \tag{34}$$

**Conclusion.** By (31)–(34), mean-square stability is controlled by $\rho(T_{\text{slow}}(\eta, \beta))$. In the noise-dominated regime (small batch) where the multiplicative term dominates the remainder, i.e. $\eta_{\text{eff}}^2 \|G\| \gg \|R(\eta, \beta)\|$, we obtain the advertised condition

$$\rho\big(I_{d^2} - \eta_{\text{eff}} K + \eta_{\text{eff}}^2 G\big) < 1,$$

which matches the SGD stability operator of Ma & Ying (2021, Eq. (31)) with step size $\eta_{\text{eff}}$. $\qquad\square$

# F. Interventions Mid-Training

We present additional intervention experiments that complement our main points by illustrating the robustness of the batch sharpness response.

## F.1. Destabilizing Intervention

We first consider interventions in which a single hyperparameter is modified mid-training so as to lower the effective stability threshold (by increasing the learning rate, increasing the momentum, or decreasing the batch size). Figure 6, Figure 8, Figure 9, and Figure 10 show destabilizing intervention experiments across varying batch-size regimes and intervention timings (i.e, both during and after the progressive sharpening phase). The batch sharpness trajectory of the intervention run (red) closely follows that of the baseline run (blue) prior to the intervention and rapidly transitions to track that of the destabilized run (purple) after the intervention. When the intervention causes the effective stability threshold to drop below the current batch sharpness level, training exhibits a *catapult*: a sharp increase in loss and curvature statistics followed by restabilization near the new threshold. These results confirm that batch sharpness responds immediately and predictably to destabilizing hyperparameter changes and supports the interpretation that training dynamically tracks a hyperparameter-dependent stability boundary.

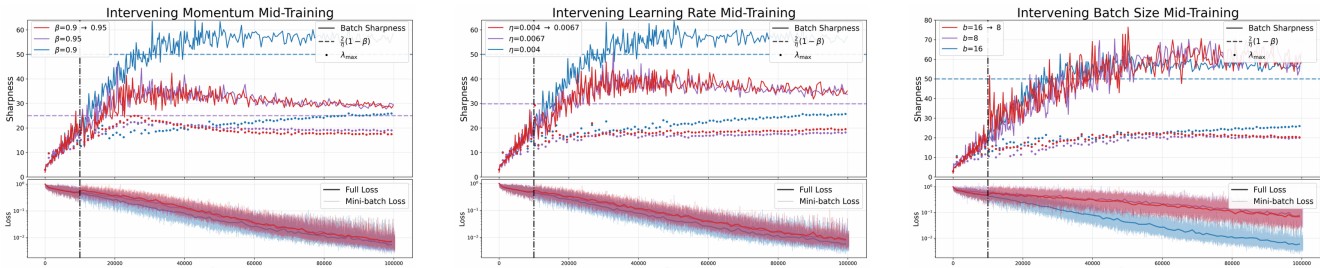

*Figure 8.* Within-run EoSS dynamics for **early destabilizing interventions** during the progressive sharpening phase at step 10k on an MLP with baseline learning rate $\eta = 0.004$, momentum $\beta = 0.9$, and batch size $b = 16$. Left: destabilizing momentum intervention, increasing $\beta$ to 0.95. Middle: destabilizing learning-rate intervention, increasing $\eta$ to 0.0067. Right: destabilizing batch-size intervention, decreasing batch size $b$ to 8. Top: Batch Sharpness and $\lambda_{\max}$. Bottom: Training loss.

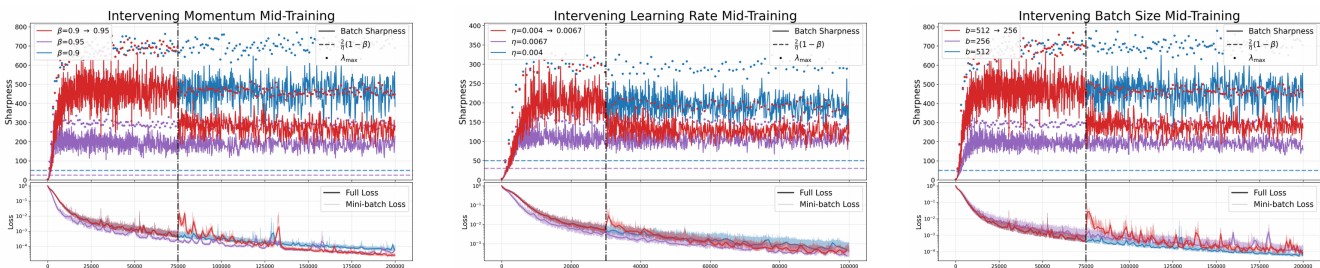

*Figure 9.* Within-run EoSS dynamics for **destabilizing interventions at an intermediate batch size** at step 30k on an MLP with baseline learning rate $\eta = 0.004$, momentum $\beta = 0.9$, and batch size $b = 512$. Left: destabilizing momentum intervention, increasing $\beta$ to 0.95. Middle: destabilizing learning-rate intervention, increasing $\eta$ to 0.0067. Right: destabilizing batch-size intervention, decreasing batch size $b$ to 256. Top: Batch Sharpness and $\lambda_{\max}$. Bottom: Training loss.

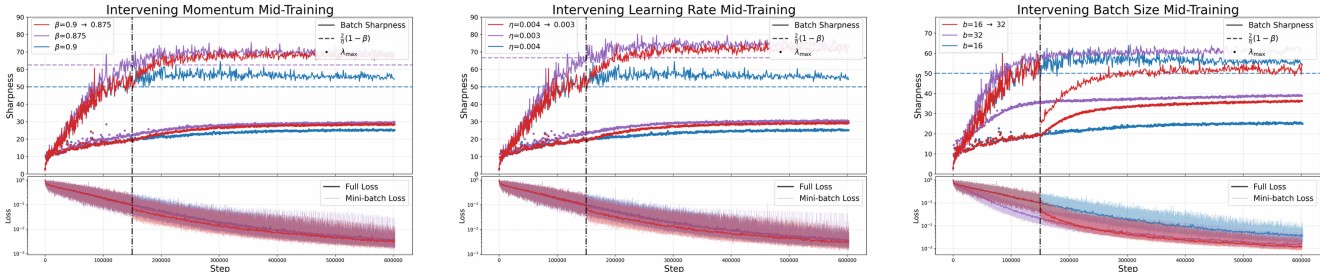

*Figure 10.* Within-run EoSS dynamics for **destabilizing interventions at high batch size** at step 50k on an MLP with baseline learning rate $\eta = 0.03$, momentum $\beta = 0.5$, and batch size $b = 16384$. Left: destabilizing momentum intervention, increasing $\beta$ to 0.52. Middle: destabilizing learning-rate intervention, increasing $\eta$ to 0.035. Right: destabilizing batch-size intervention, decreasing $b$ to 12288. Top: Batch Sharpness and $\lambda_{\max}$. Bottom: Training loss.

## F.2. Stabilizing Intervention

We next consider stabilizing interventions by decreasing the learning rate, decreasing the momentum, or increasing the batch size across varying batch sizes and timings in Figure 11, Figure 12, and Figure 13. In contrast to destabilizing interventions, stabilizing interventions induce an immediate decrease in the training loss instead of a *catapult*. In addition, the intervention modifies the long-term evolution of Batch Sharpness by permitting further progressive sharpening toward a higher plateau.

For small batches and for interventions both during and after the progressive sharpening phase, the intervention run transitions away from the baseline trajectory and gradually approaches the trajectory of the stabilized run trained from initialization with the modified hyperparameters. This indicates that raising the stability threshold reopens a sharpening phase that had previously saturated. Nevertheless, for stabilizing interventions at intermediate batch sizes (Figure 13), the batch sharpness remains close to that of the baseline run rather than approaching the stabilized run. Under this hyperparameter regime, we do not observe a continuation of progressive sharpening, as the network has already effectively converged at the time of intervention. This suggests that once the learning dynamics have saturated, raising the stability threshold alone is insufficient to reinitiate sharpening.

*Figure 11.* Within-run EoSS dynamics for **stabilizing interventions with low batch sizes** at step 150k on an MLP with batch size $b = 16$, learning rate $\eta = 0.004$, and momentum $\beta = 0.9$. Left: stabilizing momentum intervention, decreasing $\beta$ to 0.875. Middle: stabilizing learning-rate intervention, decreasing $\eta$ to 0.003. Right: stabilizing batch-size intervention, increasing batch size $b$ to 32. Top: Batch Sharpness and $\lambda_{\max}$. Bottom: Training loss.

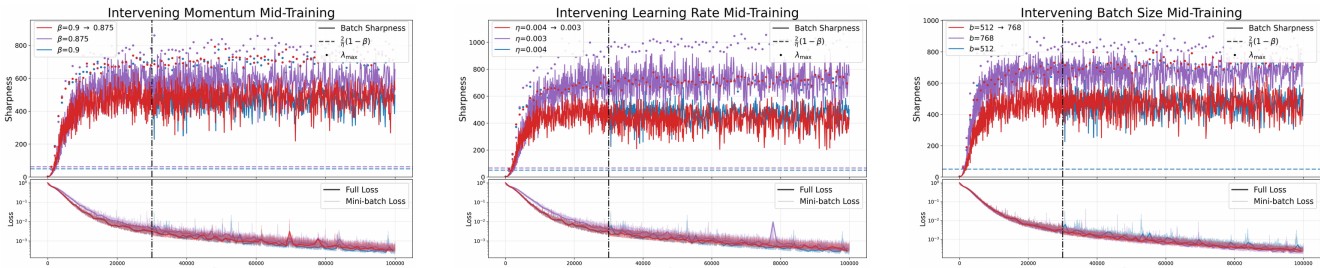

*Figure 12.* Within-run EoSS dynamics for **early stabilizing interventions** during the progressive sharpening phase at step 10k on an MLP with baseline learning rate $\eta = 0.004$, momentum $\beta = 0.9$, and batch size $b = 16$. Left: stabilizing momentum intervention, decreasing $\beta$ to 0.875. Middle: stabilizing learning-rate intervention, decreasing $\eta$ to 0.003. Right: stabilizing batch-size intervention, increasing batch size $b$ to 32. Top: Batch Sharpness and $\lambda_{\max}$. Bottom: Training loss.

*Figure 13.* Within-run EoSS dynamics for **stabilizing interventions with intermediate batch sizes** at step 75k on an MLP with baseline learning rate $\eta = 0.004$, momentum $\beta = 0.9$, and batch size $b = 512$. Left: stabilizing momentum intervention, decreasing $\beta$ to 0.875. Middle: stabilizing learning-rate intervention, decreasing $\eta$ to 0.003. Right: stabilizing batch-size intervention, increasing batch size $b$ to 768. Top: Batch Sharpness and $\lambda_{\max}$. Bottom: Training loss.

## G. Distance in the Parameter Space

In Figure 14, we evaluate whether SGDM and vanilla SGD with matching stabilization levels of Batch Sharpness and $\lambda_{\max}$ (as in Figure 5) follow comparable parameter trajectories. To compute the $L_2$ distance in the left plot, we apply a fixed JL projection to reduce the weights to a 5,000-dimensional subspace, allowing us to compute trajectory distances efficiently while maintaining geometric fidelity. On the right plot, we use the notion of a test prediction distance, defined as the Frobenius norm between the networks' output logits on CIFAR-10's held-out test set of size 10,000. In addition to measuring distance at matching training steps, we introduce the notion of *true distance*: the minimum distance from each SGD step to any point along the entire SGDM trajectory. This metric is motivated by the observation that even if two trajectories diverge at matching step counts, they may still traverse the same regions of parameter space at different rates.

Figure 14 uses the distance from initialization primarily as a baseline to provide context for the separation between SGD and SGDM trajectories. While both runs move a similar total distance through parameter space, the distance between them is of a comparable order of magnitude to their distance from initialization. This lack of point-by-point proximity suggests that matching Batch Sharpness stabilization levels does not ensure a strong approximation between the two methods.

This divergence is equally present in function space, where test prediction distances indicate that the networks learn meaningfully different input-output mappings. These results confirm that SGD and SGDM explore geometrically distinct regions of the landscape, though this observation does not rule out the possibility of a weak approximation (as in Wang et al. (2024)) in which the statistical properties of the trajectories might still align.

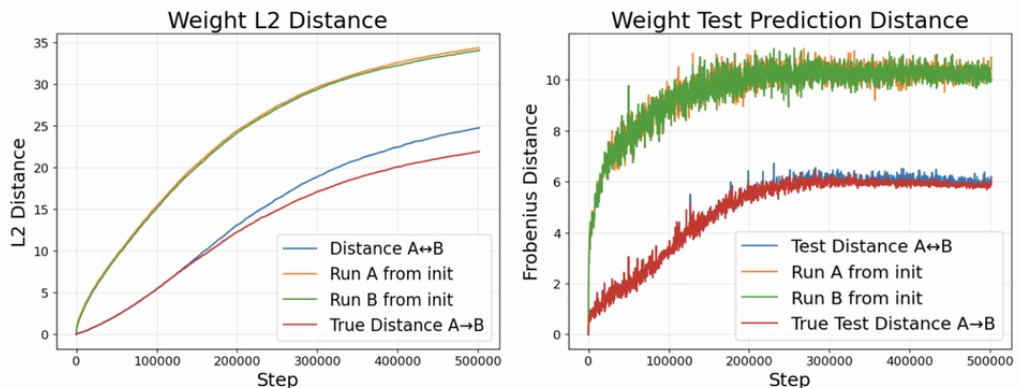

*Figure 14.* Distance metrics comparing training trajectories of SGDM$(\eta, \beta, b)$ and SGD$(\frac{\eta}{1-\beta}, b)$ for $\eta = 0.001$, $\beta = 0.9$, and $b = 4$. The hyperparameters are chosen as in Figure 5, so that Batch Sharpness stabilizes at 200 for both runs. Each panel shows four curves: the distance of the SGD weights from initialization (orange), the distance of the SGDM weights from initialization (green), the distance between the two runs at matching steps (red), and the true distance between the two trajectories. The left plot uses $L_2$ distance in a weight space projected down to 5,000 dimensions. The right plot uses test prediction distance, measuring functional rather than weight-space divergence.

## H. Additional Adam Details

Cohen et al. (2022) analyze Adam through a "frozen Adam" approximation: at a checkpoint, the diagonal preconditioner $P_t$ is held fixed, reducing the short-term dynamics to preconditioned heavy-ball momentum. On a local quadratic with Hessian $H_t$, this makes stability depend on the preconditioned Hessian $P_t^{-1/2} H_t P_t^{-1/2}$, equivalently on the spectrum of $P_t^{-1} H_t$. For EMA-style heavy-ball momentum, $m_{t+1} = \beta_1 m_t + (1 - \beta_1) g_t$, the quadratic stability threshold is $2(1 + \beta_1)/((1 - \beta_1)\eta)$; the denominator is absent in our SGDM convention (5) because that update accumulates unnormalized gradients. This explains why the deterministic Adam threshold is the SGDM full-batch threshold after replacing raw sharpness by preconditioned sharpness and accounting for EMA normalization.

For small-batch Adam, the corresponding stochastic criterion is unknown. The heuristic in Section 5.1 is that, if the adaptive dynamics are noise-dominated, the SGDM small-batch factor $(1 - \beta_1)$ appears in the numerator. Because Adam uses EMA-style momentum, this factor cancels the denominator $(1 - \beta_1)$ in the deterministic Adam threshold, yielding an approximately $\beta_1$-independent preconditioned-sharpness level. Our experiments use preconditioned $\lambda_{\max}$ only as a proxy for this cancellation effect. Establishing the right adaptive analogue of Batch Sharpness remains open.

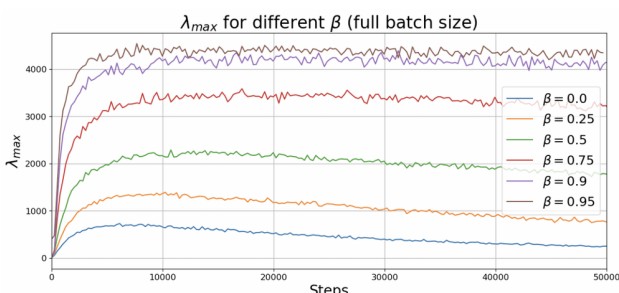 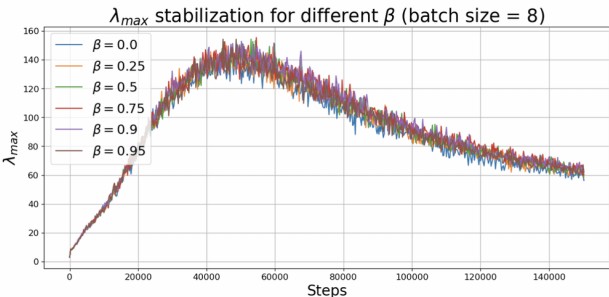

*Figure 15.* **Raw sharpness for Adam across momentum.** Left: in full-batch Adam, the raw $\lambda_{\max}$ reaches higher values as $\beta_1$ increases. Right: at small batch size ($b = 8$), the raw $\lambda_{\max}$ trajectories are much lower and nearly independent of $\beta_1$. These trends are consistent with the two-regime picture in Section 5.1, but raw $\lambda_{\max}$ is only a supporting diagnostic for Adam: as Cohen et al. (2022) emphasize, adaptive stability is governed by preconditioned sharpness rather than raw sharpness.

## I. Stabilization vs Batch Size Ablations

This section shows how curvature-related quantities stabilize as a function of batch size for a fixed total sample budget for training runs on a CNN and MLP. For each optimizer and learning-rate configuration, we plot the plateau values of Batch Sharpness and $\lambda_{\max}$ obtained from within-run dynamics across increasing batch sizes. These plots illustrate the transition from the small-batch regime to the large-batch regime, and we overlay the corresponding theoretical stability thresholds implied by momentum and learning rate. Together, they make explicit how optimizer choice reshapes the location and sharpness of this transition; in particular, Nesterov typically reaches the critical batch size at much smaller batch sizes than Polyak momentum, consistent with its effectively reduced stability threshold.

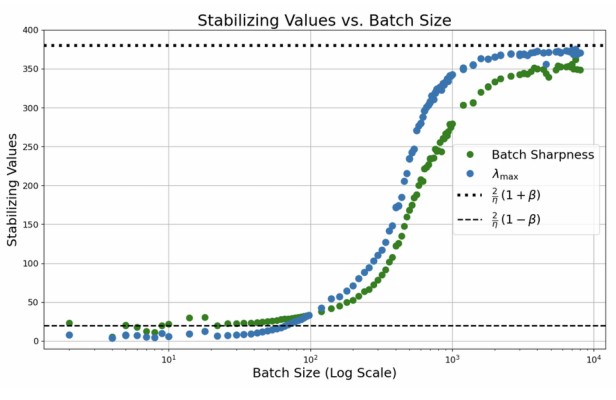

*Figure 16.* MLP, $\eta = 0.01$, $\beta = 0.9$

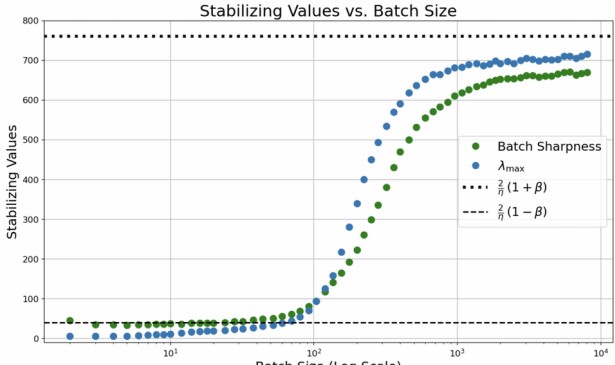

*Figure 17.* MLP, $\eta = 0.005$, $\beta = 0.9$

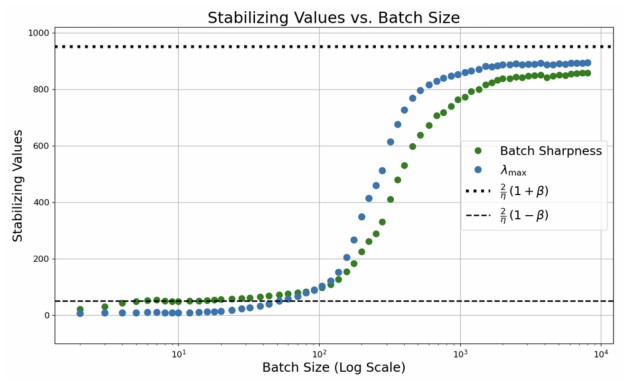

*Figure 18.* MLP, $\eta = 0.004$, $\beta = 0.9$

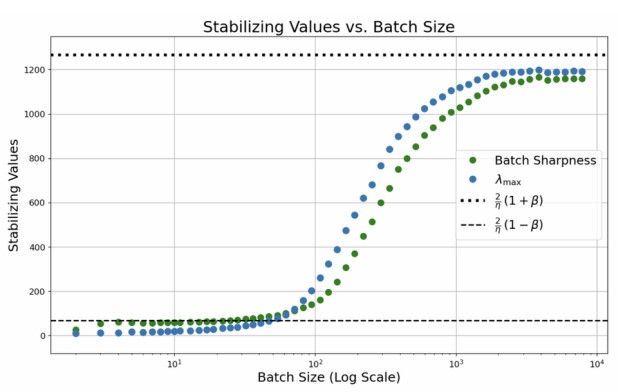

*Figure 19.* MLP, $\eta = 0.003$, $\beta = 0.9$

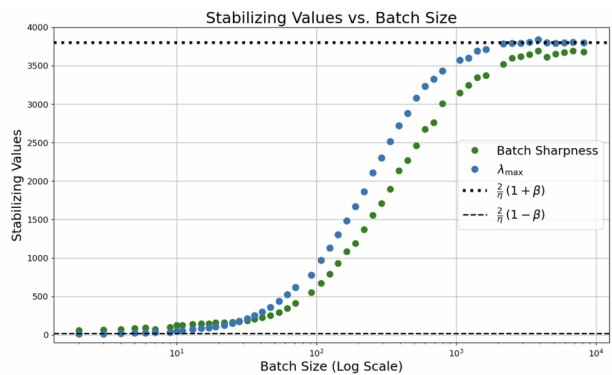

*Figure 20.* MLP, $\eta = 0.001$, $\beta = 0.9$

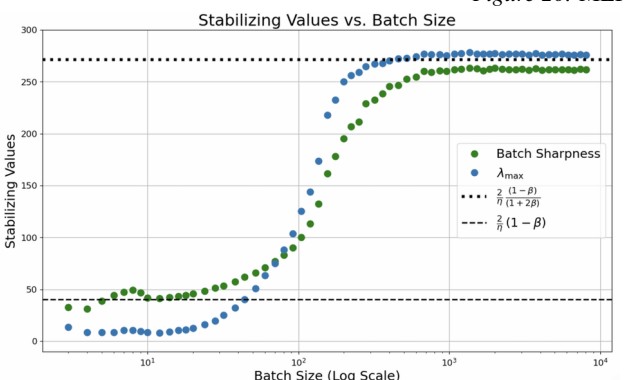

*Figure 21.* MLP, $\eta = 0.005$, $\beta = 0.9$, Nesterov

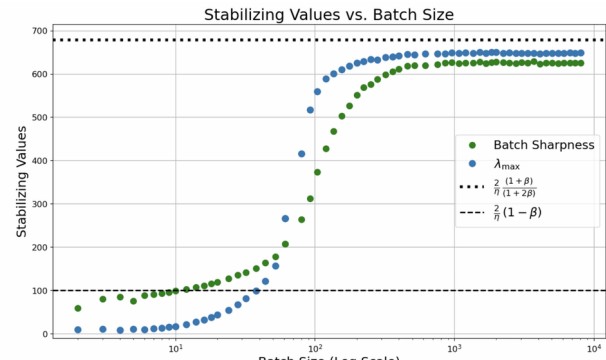

*Figure 22.* MLP, $\eta = 0.002$, $\beta = 0.9$, Nesterov

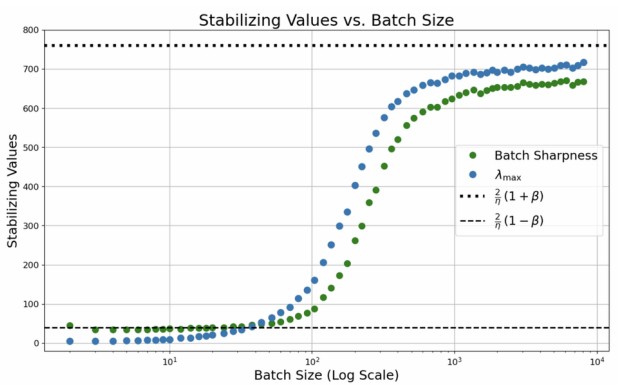

*Figure 23.* CNN, $\eta = 0.005$, $\beta = 0.9$

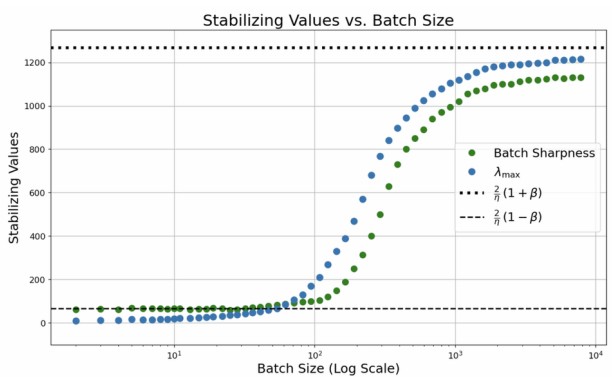

*Figure 24.* CNN, $\eta = 0.003$, $\beta = 0.9$

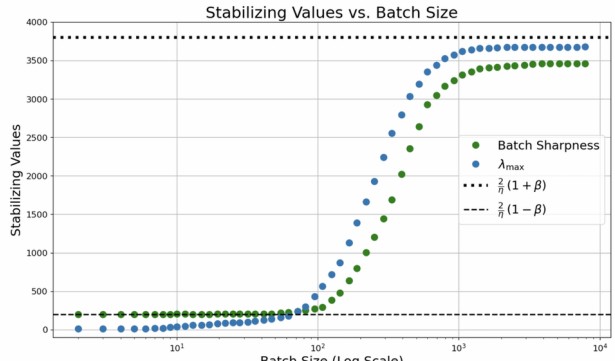

*Figure 25.* CNN, $\eta = 0.001$, $\beta = 0.9$

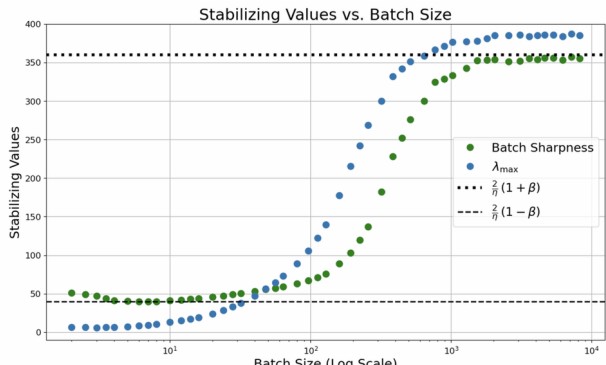

*Figure 26.* CNN, $\eta = 0.01$, $\beta = 0.8$

# J. Further Experiments

This appendix reports additional within-run dynamics grouped by ablation family. For each configuration we show batch sharpness, $\lambda_{\max}$, and training loss as a function of optimization steps, with subfigures corresponding to different batch sizes.

Unless otherwise noted, all within-run dynamics are shown for batch sizes $B \in \{2, 4, 6, 8, 16, 32, 64, 128, 256, 8192\}$. Notice how for smaller batch sizes we do not always have convergence—as illustrated in the flatness of the loss. This happens due to excessive noise for selected batch and step size.

## J.1. Momentum Ablations (MLP, $\eta = 0.01$)

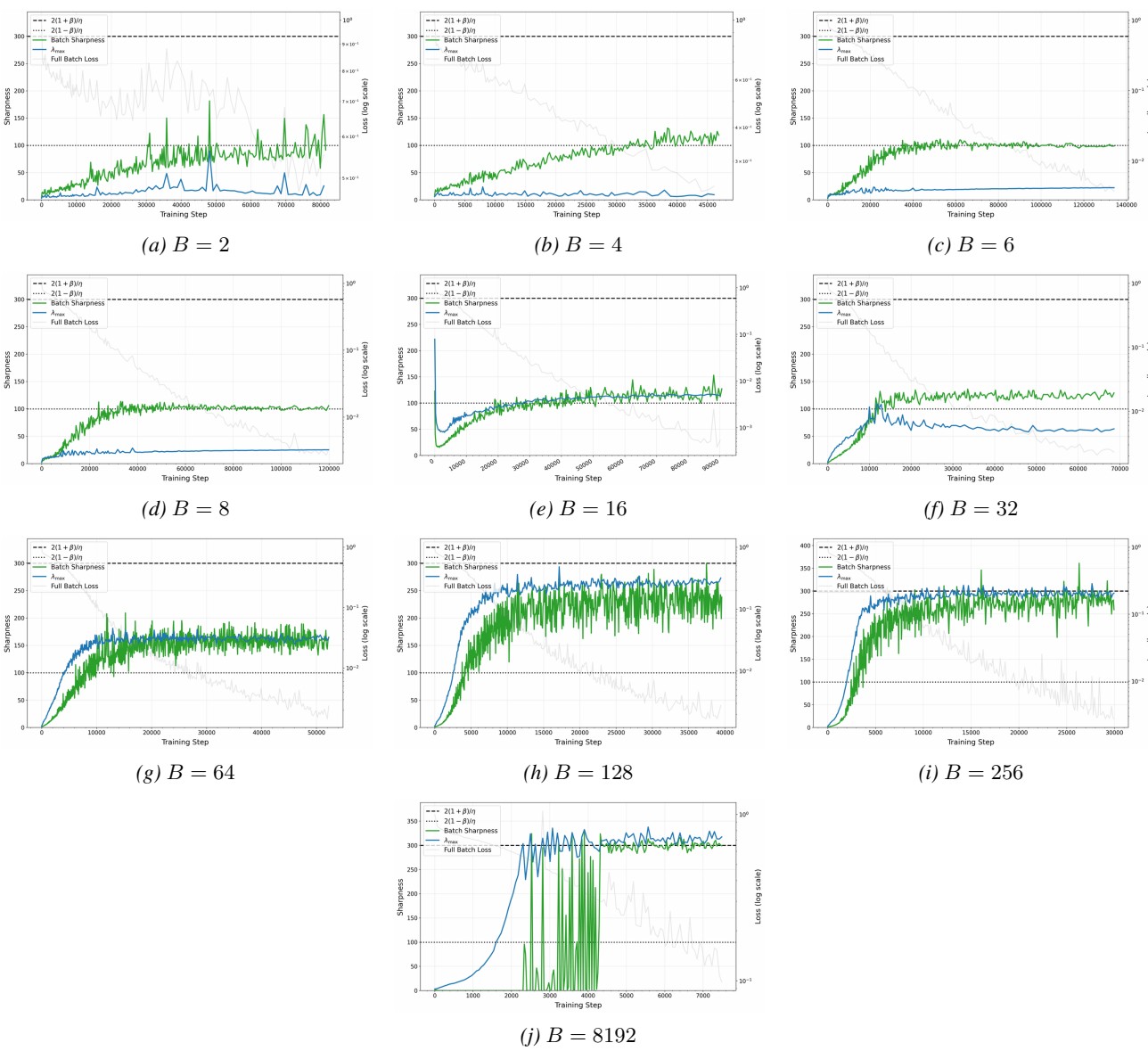

*Figure 27.* Within-run dynamics for an MLP trained with $\eta = 0.01$ and momentum $\beta = 0.5$ across batch sizes.

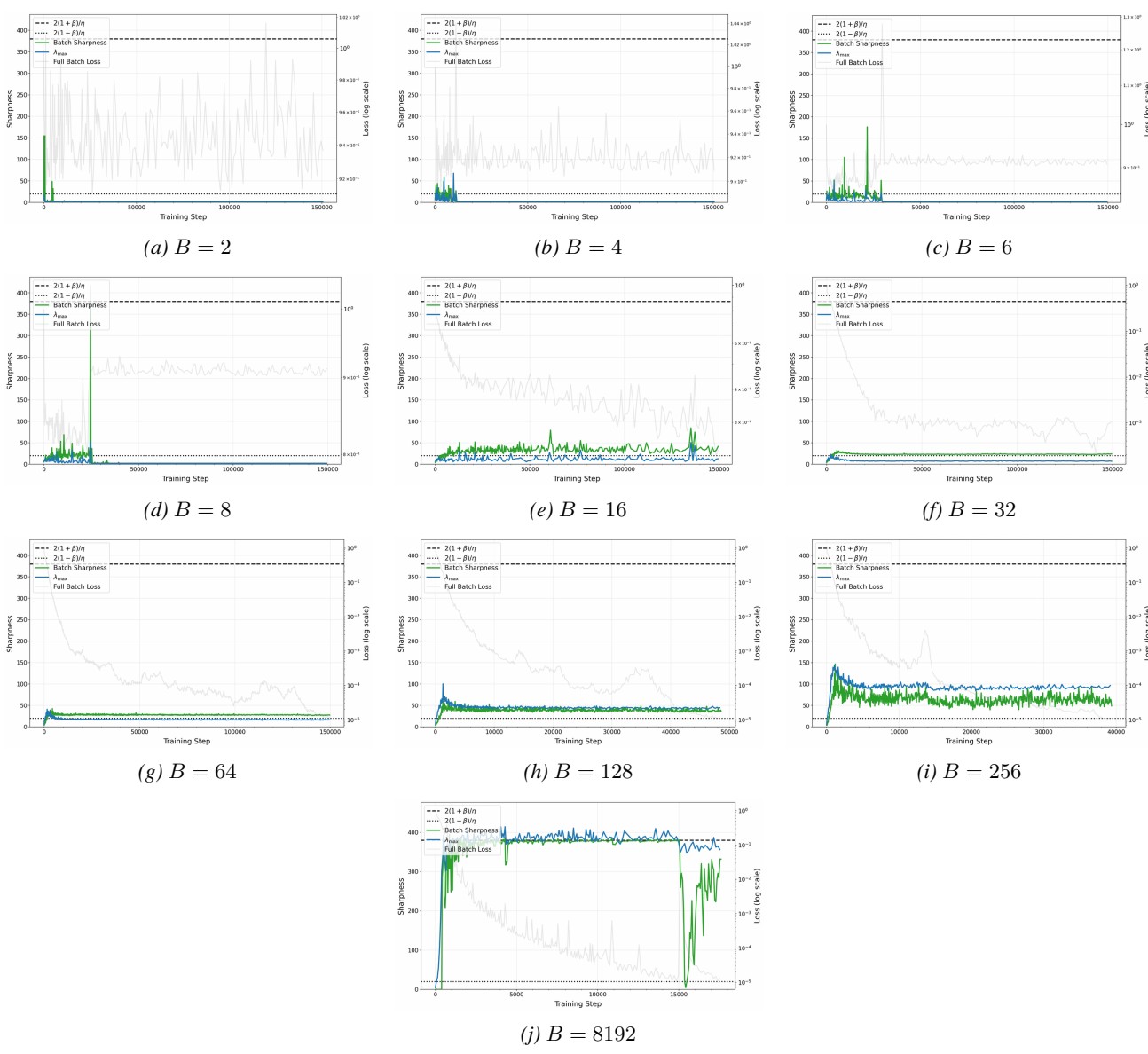

*Figure 28.* Within-run dynamics for MLP with $\eta = 0.01$ and momentum $\beta = 0.9$ across batch sizes.

### J.1.1. NOTE ON SMALLNESS OF $\lambda_{\max}$

As a concrete illustration, consider an MLP trained with learning rate $\eta = 0.01$, momentum $\beta = 0.9$, and batch size $B = 16$. For this run, the final observed curvature satisfies $\lambda_{\max} \approx 12.76$. Evaluating quantity $\frac{\lambda_{\max}^2 \eta^2}{(1-\beta)^3}$ yields a value of approximately 16.28.

## J.2. Learning-Rate Ablations (MLP, $\beta = 0.5$)

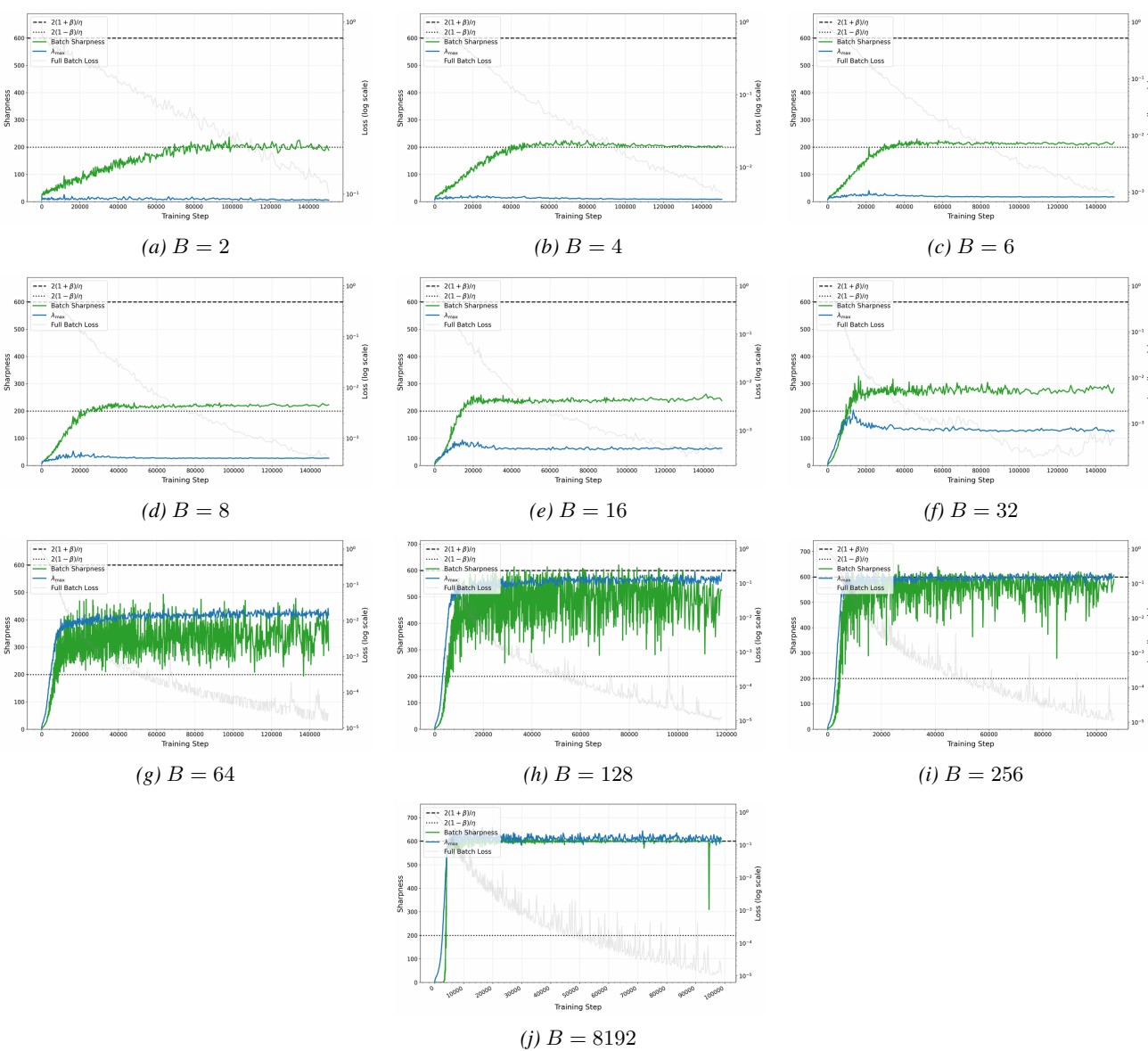

*Figure 29.* Within-run dynamics for MLP with $\eta = 0.005$ and momentum $\beta = 0.5$ across batch sizes.

## J.3. Architecture Ablations

CNN: SiLU, $\eta = 0.01$, $\beta = 0.5$

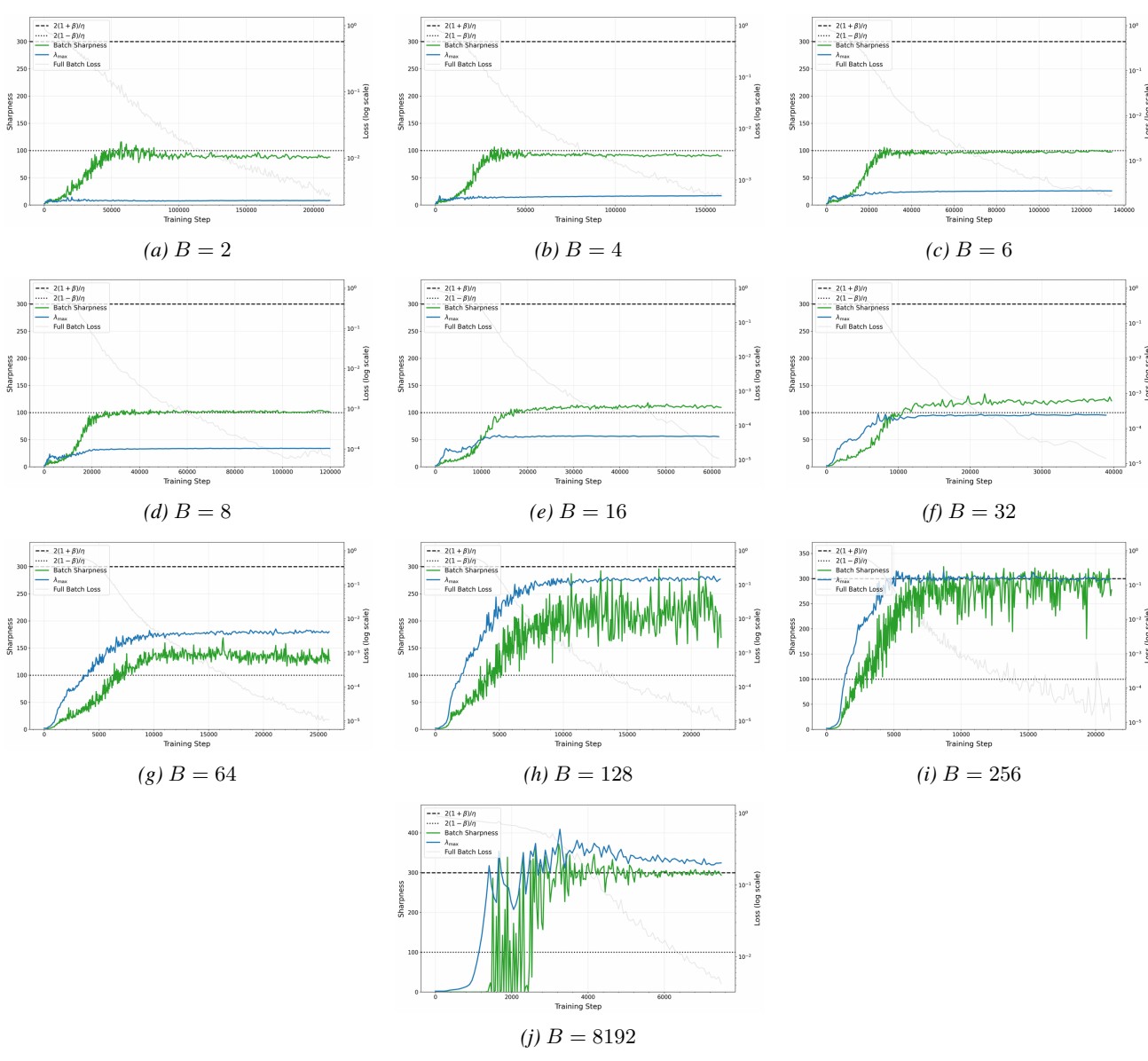

*Figure 30.* Within-run dynamics for CNN+SiLU with $\eta = 0.01$ and $\beta = 0.5$.

CNN: SiLU, $\eta = 0.01$, $\beta = 0.8$

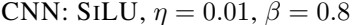

*(a)* $B = 2$

*(b)* $B = 4$

*(c)* $B = 6$

*(d)* $B = 8$

*(e)* $B = 16$

*(f)* $B = 32$

*(g)* $B = 64$

*(h)* $B = 128$

*(i)* $B = 256$

*(j)* $B = 8192$

*Figure 31.* Within-run dynamics for CNN+SiLU with $\eta = 0.01$ and $\beta = 0.8$.

CNN: SiLU, $\eta = 0.02$, $\beta = 0.8$

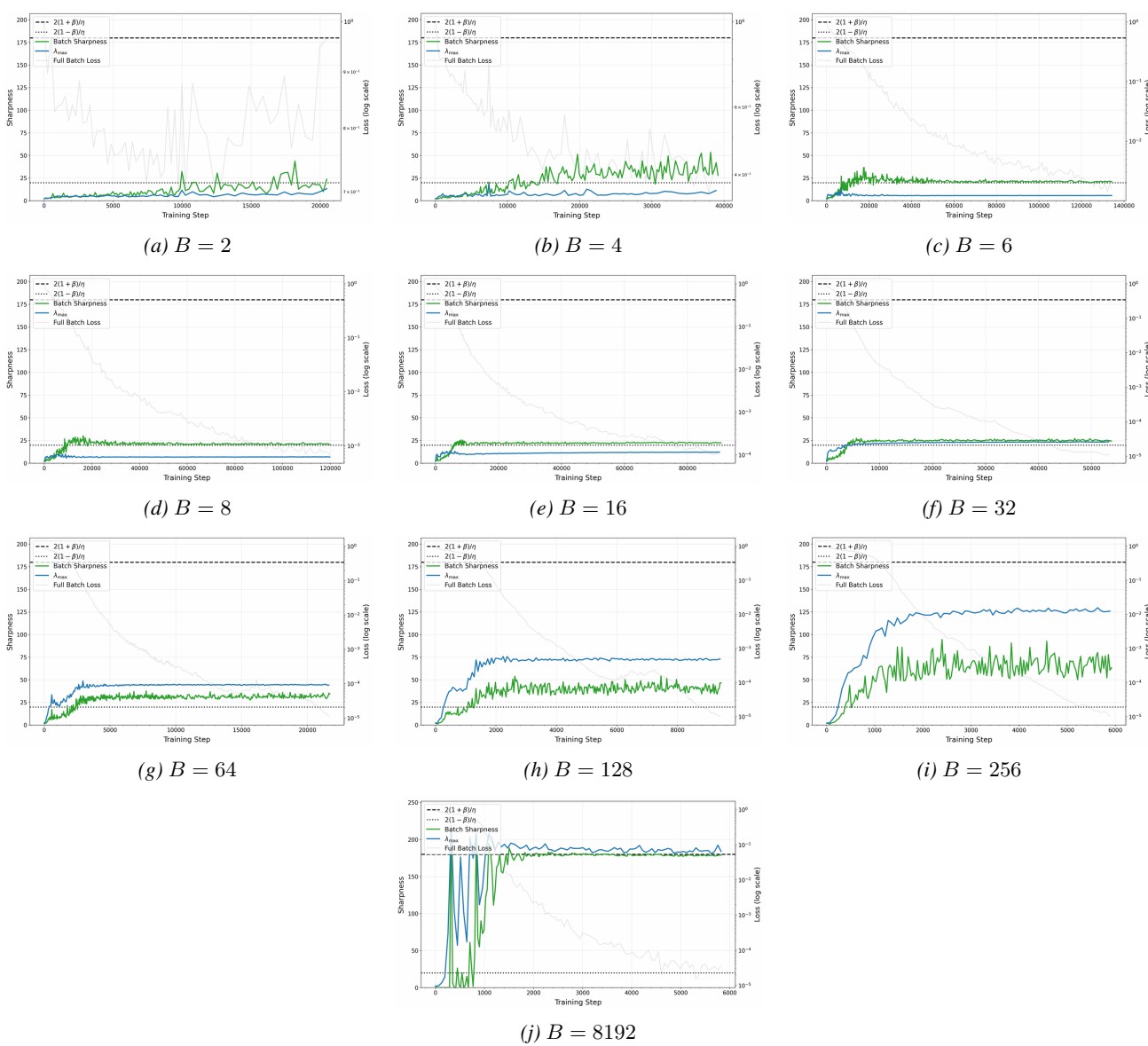

*(a)* $B = 2$

*(b)* $B = 4$

*(c)* $B = 6$

*(d)* $B = 8$

*(e)* $B = 16$

*(f)* $B = 32$

*(g)* $B = 64$

*(h)* $B = 128$

*(i)* $B = 256$

*(j)* $B = 8192$

*Figure 32.* Within-run dynamics for CNN+SiLU with $\eta = 0.02$ and $\beta = 0.8$.

RESNET: RELU, $\beta = 0.5$, $\eta = 0.02$

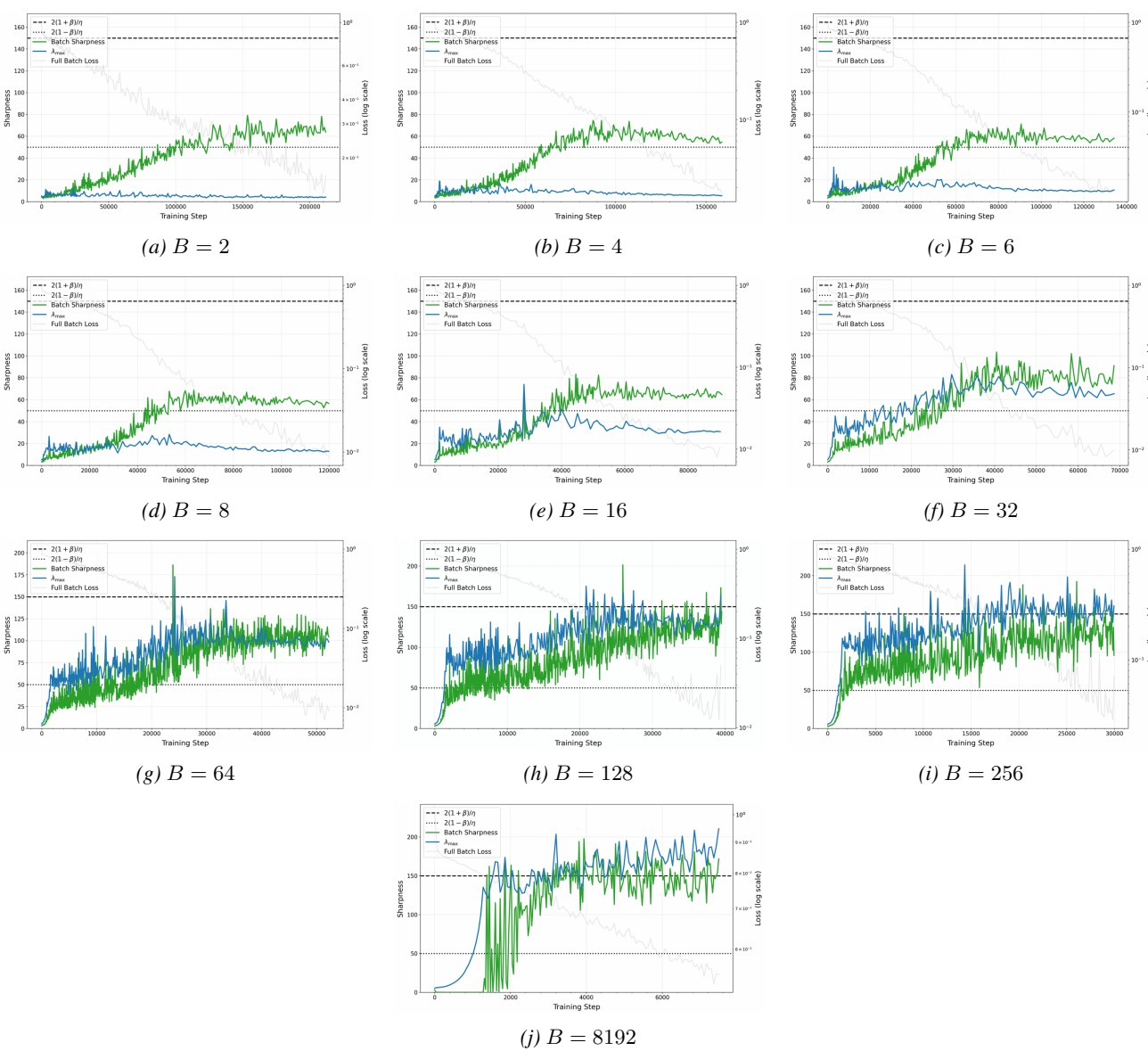

*Figure 33.* Within-run dynamics for ResNet+ReLU with $\eta = 0.02$ and $\beta = 0.5$.

## J.4. Activation Ablations

This section isolates activation function effects in an MLP at fixed optimizer settings: $\beta = 0.5$ and $\eta = 0.01$.

MLP: RELU, $\beta = 0.5$, $\eta = 0.01$

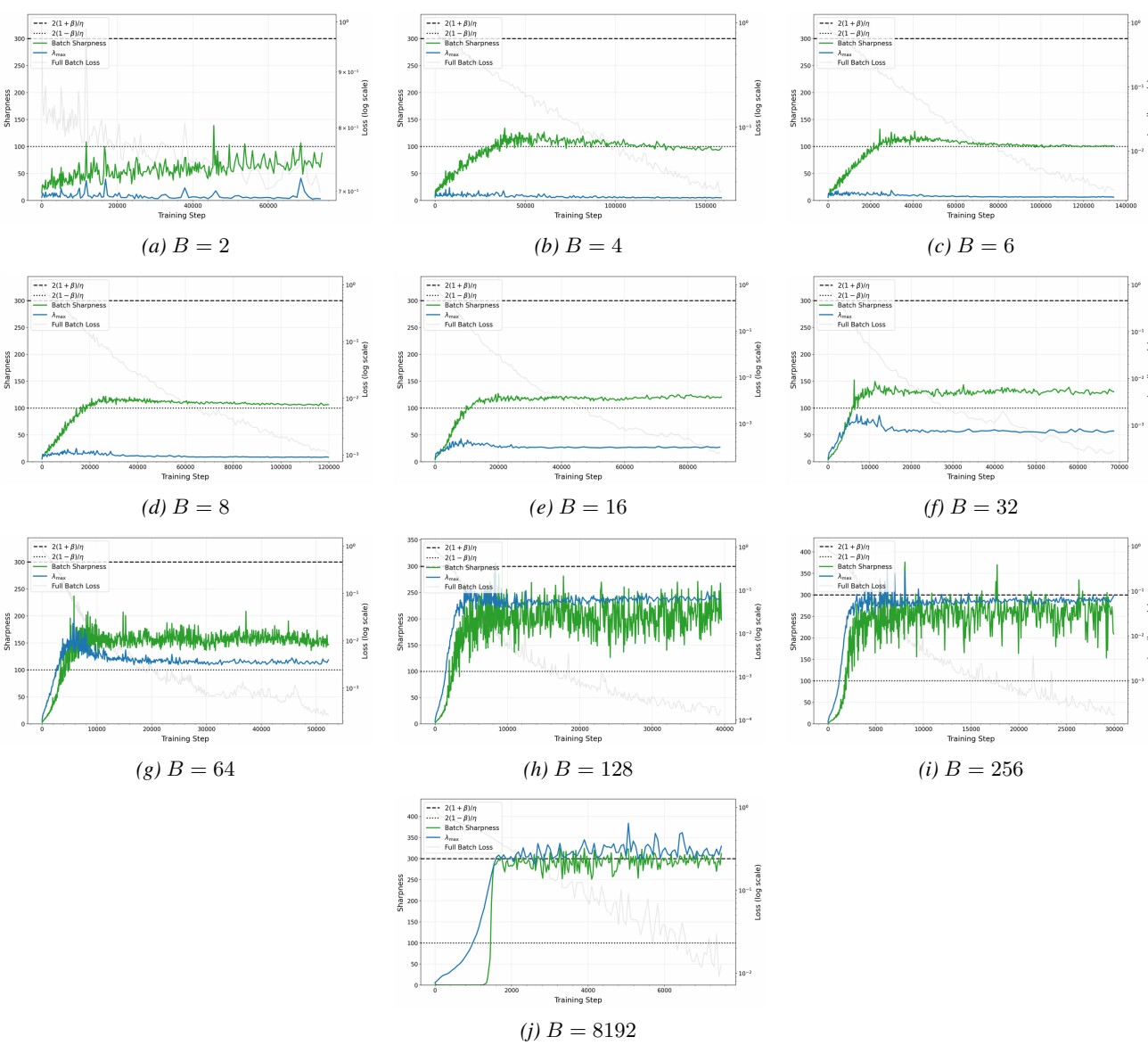

*Figure 34.* Within-run dynamics for ReLU MLP ($\eta = 0.01$, $\beta = 0.5$) across batch sizes.

MLP: SILU, $\beta = 0.5$, $\eta = 0.01$

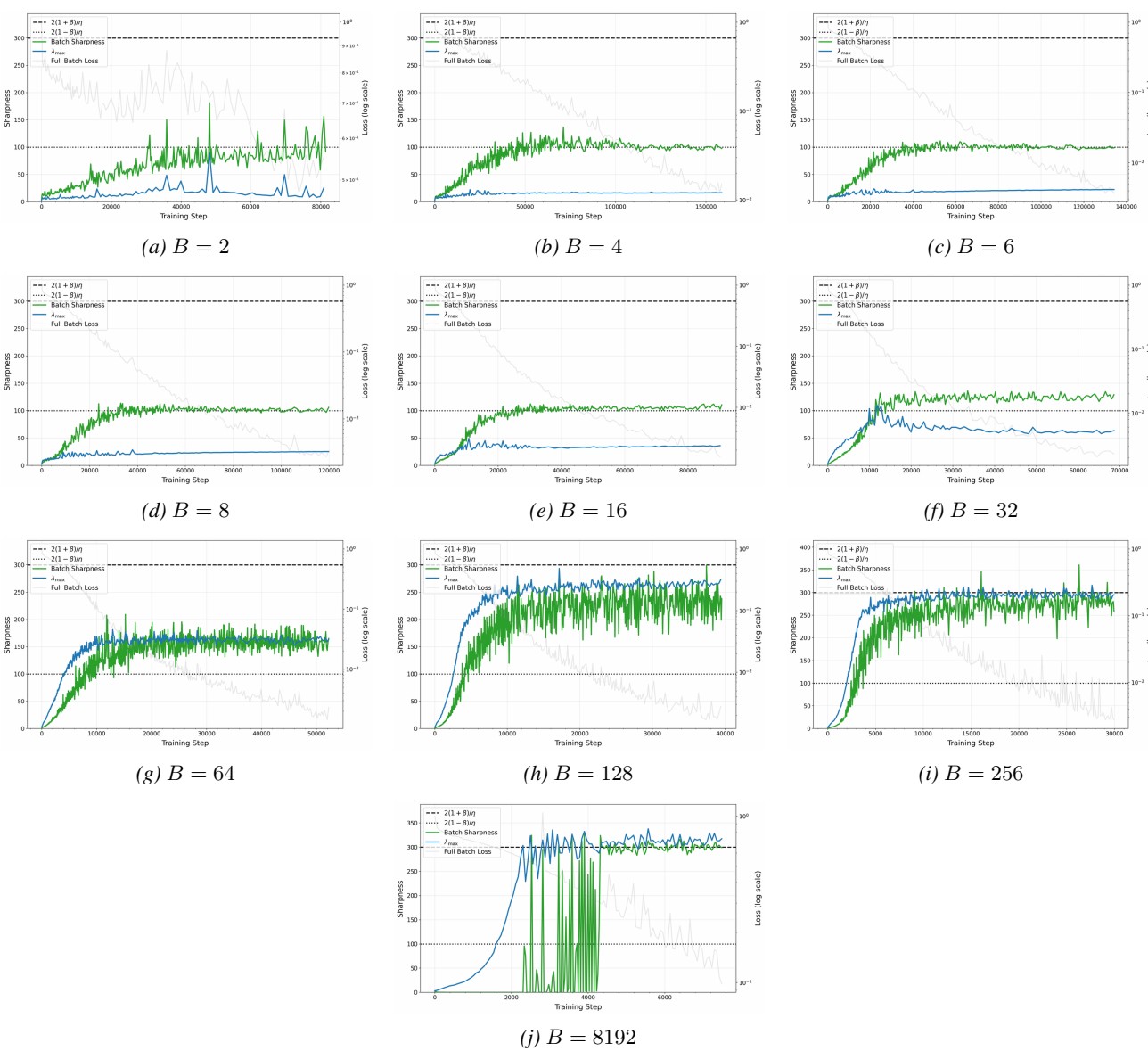

*Figure 35.* Within-run dynamics for SiLU MLP ($\eta = 0.01$, $\beta = 0.5$) across batch sizes.

VɪT, $\beta = 0.5$, $\eta = 0.007$

The full-batch ViT plots should be read with one architecture-specific caveat. In these runs, $\lambda_{\max}$ can stabilize well above the nominal GD-M threshold $2/\eta(1 + \beta)$. Ghosh et al. (2025) report the same above-threshold sharpness behavior for ViTs and associate it with attention-entropy collapse. Zhai et al. (2023) identified this collapse as a Transformer instability mode in which low attention entropy is accompanied by high sharpness and loss instability. Thus, for ViTs, the full-batch $\lambda_{\max}$ stabilization level may include attention-specific effects beyond the batch-size-dependent momentum picture studied in the main text. Note that Batch Sharpness still stabilizes at the exact level of $2/\eta(1 + \beta)$.

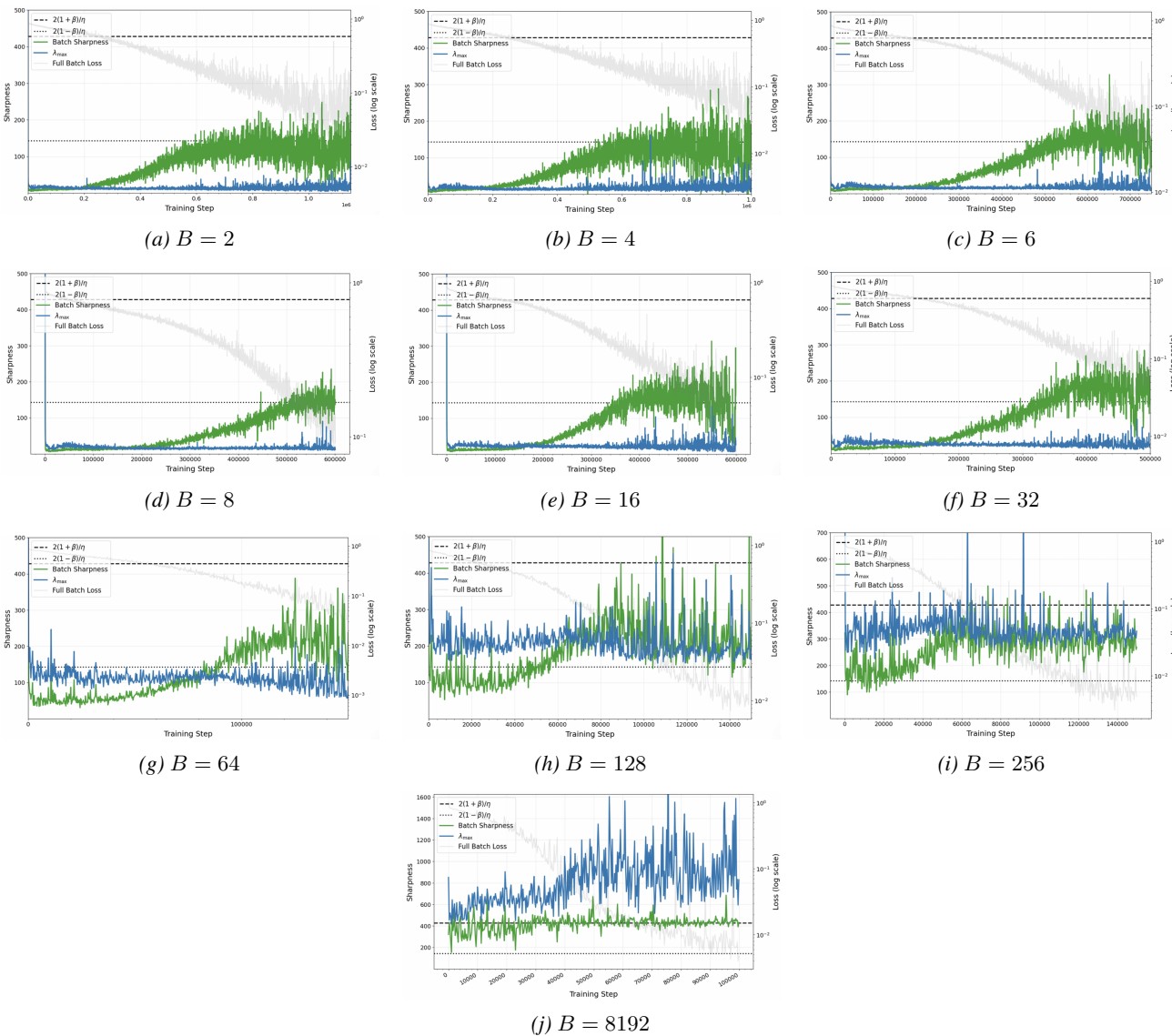

*Figure 36.* Within-run dynamics for a ViT ($\eta = 0.007$, $\beta = 0.5$) across batch sizes using the MSE loss.

CNN, $\beta = 0.9$, $\eta = 0.001$

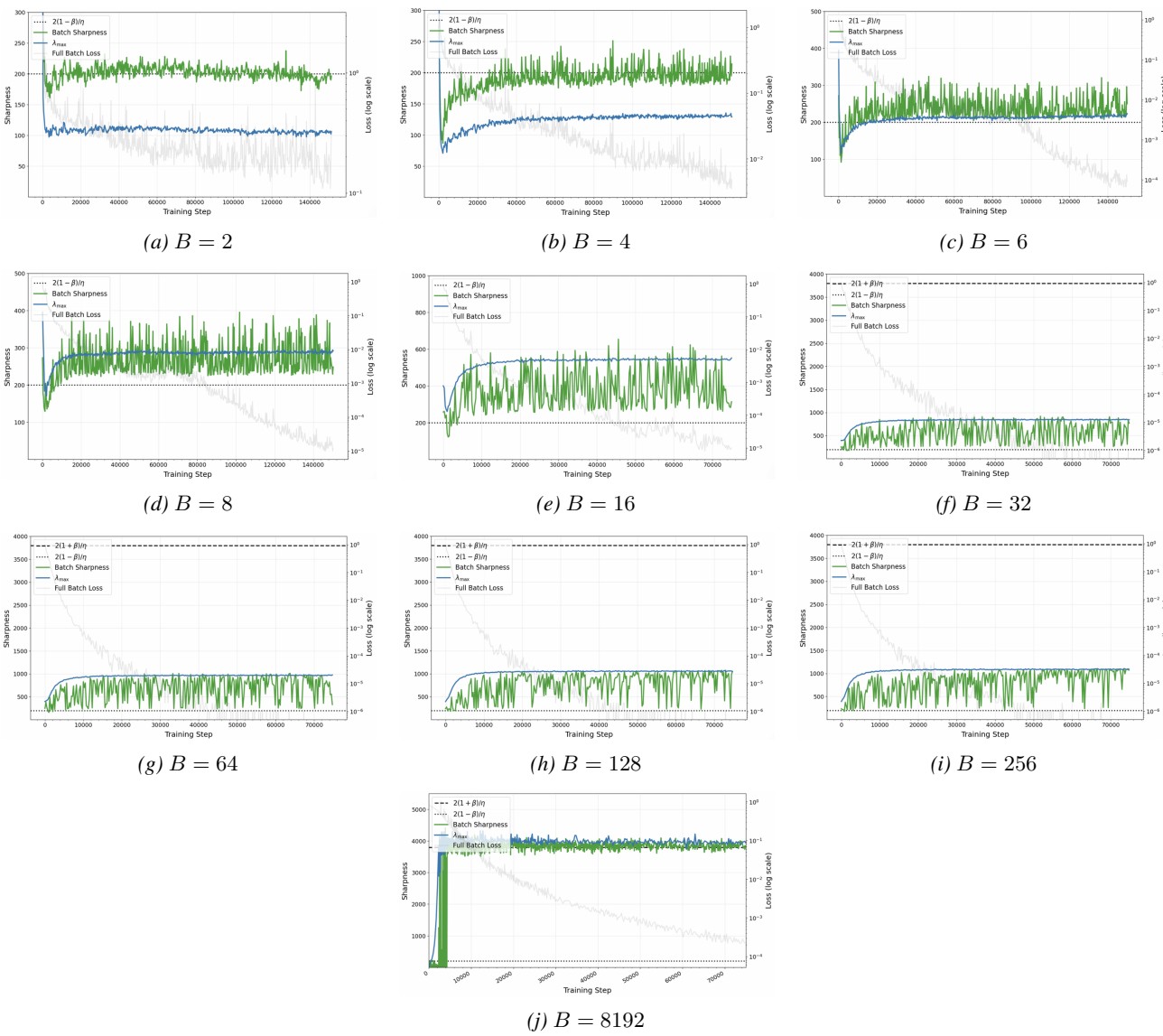

*Figure 37.* Within-run dynamics for a CNN ($\eta = 0.001$, $\beta = 0.9$) trained on the SST dataset across batch sizes using the MSE loss.

MLP, $\beta = 0.75$, $\eta = 0.008$

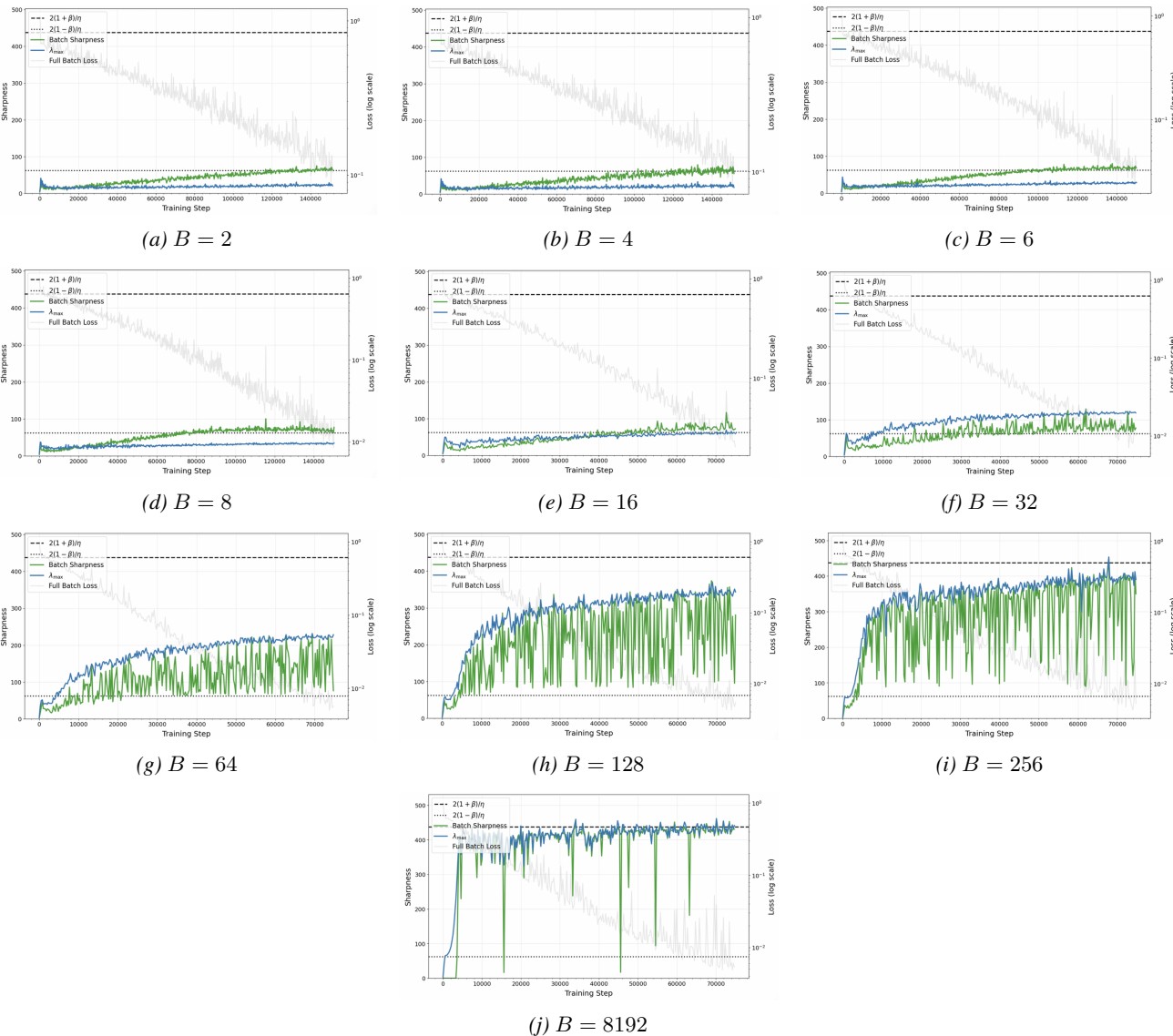

*Figure 38.* Within-run dynamics for an MLP ($\eta = 0.008$, $\beta = 0.75$) trained on the SST dataset across batch sizes using the MSE loss.

