# OpenReview forum: "Momentum Further Constrains Sharpness at the Edge of Stochastic Stability"
_ICML.cc/2026/Conference — ICML 2026 regular_

### Official Review · Reviewer_nDAU · 2026-02-23

**Soundness:** 2
**Presentation:** 1
**Significance:** 3
**Originality:** 3
**Overall Recommendation:** 4
**Confidence:** 4

**Summary:**

The paper investigates the concept of training near the Edge of Stability for mini-batch SGD with momentum (Polyak's and Nesterov's), which has been investigated separately ([1] for mini-batch SGD, [2] for GD with momentum), but not jointly. The findings are mainly empirical, and aim to show that the stability boundary exhibit two regimes with respect to batch size: small batch (noise dominated) regime, with a smaller instability threshold (enforcing to stay in regions with smaller curvature, so flatter), and large batch regime, closer to the deterministic setting. There is an intermediate regime between these two. On the theoretical side, a stability analysis is performed on quadratic, which supports the stability bound observed by the authors, while not being strictly an stability bound in a similar way than the usual EoS litterature, as crossing this bound does not make the algorithm diverge.

[1] Revisiting the Edge of Stability for SGD, Andreyev, A. and Beneventano, P. Edge of Stochastic Stability

[2] Gradient Descent on Neural Networks Typically Occurs at the Edge of Stability, Jeremy M. Cohen, Simran Kaur, Yuanzhi Li, J. Zico Kolter, Ameet Talwalkar

**Compliance With Llm Reviewing Policy:**

Affirmed.

**Final Justification:**

Initially, I had important concerns about some overclaims or about the presentation. The authors were receptive and took my comments on board, even though I cannot read the new version. Provided these clarification are made in the article, I think it is an interesting contribution.

**Key Questions For Authors:**

- In regard with my previously mentioned weakness (ii), how can your bound be assessed in the small batch regimes ? How to be sure we are not just interpolated between the large batch bound and a actually smaller bound ?


- Why batch sharpness is not the batch-wise sharpness, or its expectation ? This would make it reduce to sharpness in the full batch case.

**Limitations:**

yes

**Strengths And Weaknesses:**

**Strengths**
I think the subject of investigating EoS when combining mini-batch and momentum, is both interesting and challenging. The finding of the two different batch-size regime is interesting, and also in particular the fact that in small-batch regime, momentum enforces to stay in flatter regions, inducing that momentum+mini-batch creates a specific implicit bias.

**Weaknesses**
Despite interesting subject and ideas, I think the paper is significantly far from ready to be accepted. The main weaknesses are flaws in the presentation making the paper too often exceedingly difficult to read, too much statements that are not enough supported and overall too weak empirical evaluation.

**Main weaknesses**

*(i) Writing and clarity.* Overall, I found the paper hard to read. This mainly due to the massive use of vague and/or dense wording, unexplained or unintroduced formulations. I provide targeted details:
   1. About Section 2. There are too much vague wording and vague statements in this section. Examples include: "can be compared to SGD via an effective step size" compared in which sense ? "with closely tracking trajectories" ? "EOS vs. mini-batch: why the signal breaks." which signal ? "with distinct transient exploration properties"  distinct from what ? More broadly, many sentences are difficult to understand if you are not very familiar with the related works. An illustrative example is the sentence "temporal filtering and noise-shaping, since the momentum buffer is an exponential moving average of stochastic gradients, motivating SDE/stationary and modified-equation analyses". To a non-expert reader, this sentence actually does not provide any insightful explanation: it is way too dense. If you want to explain it, you should take the necessary space. I think saving space is possible; for instance by deferring parts such as "systems/implementation interactions" in the appendix, as it has not much in common with your work apart that it is about momentum.


 2. About Section 3. Overall I think Section 3 suffers from the same writting flaws than Section 2, making it hard to read and understand. Section 3 is introduced with "saturation of a computable one-sided instability certificate". What do you mean by "one-sided" ? I find this statement neither detailed nor intuitive. "whereas stabilizing perturbations do not." the concept of "stabilizing perturbations" is not introduced. "perturb the step size in the opposite direction" in the opposite of which direction ? "We will further test instability criteria (Definition A.1) found to govern the instability of optimizers without momentum or acceleration to understand the level they saturate with respect to those" I don't understand this sentence. Next paragraph, "the instability-centric viewpoint" is another example of vague wording, and I don't understand what you mean. I note that section 3.1 and 3.2 are rather clear, apart from the fact that the notations "SGDM (η, β, b)$ and SGD( η/(1−β) , b)" are not defined, and I guess "(and the flips)" should be "(and then flips)". Section 3.3 focuses on the concept of "instability boundary". You talk about how to diagnosis it, but you don't really define it, such that what you mean seems ambiguous to me. Do you mean it as a divergence criterion in the quadratic case ? "Following the discussion in Section 2, the definitive diagnostic for such a regime is..." given the importance of the statement, I think you should be more clear and detailed here and not just refer to section 2. "mid-training" is fairly vague, what is the criterion for being in "mid-training" ?

3. About Section 4 and 5. The last sentence of Section 4.2 seems very unclear, in particular "operator-centric", "to leading order", or what you mean with "due to explicit condition on ¯H operator in the constraint". On page 7: "The relevant finite-η comparator" I do not understand what it means. "the operative plateau level" what do you mean by "operative"?

To sum up about the writing and clarity, I think it is significantly below the ICML standards.

*(ii) About the main claim.*

 *1. On the relevance of the mini-batch case bound.* There are several cases where the appearing batch sharpness is below your predicted bound for small batches (Figure A1, Figure A4, Figure A5). In the beginning of Section H, you explain that for small batches the algorithm may not converge. This implies that in this case batch sharpness is not informative, and we should wait the algorithm to be convergent. But it is not that clear in which case the algorithm is converging, and as a consequence in which case the "batch sharpness" curve is informative. For instance, on Figure A1 (a), the batch sharpness is clearly below your predicted bound, and looking at the loss curve I would not say that it is not converging. Together with the fact that your linear stability analysis is weaker that usual, not ensuring divergence, makes the relevance of this bound less convincing. There is no discussion about this in the paper.
 2. About the "monotonic trend of batch size". Sometimes the batch sharpness get lower while increasing batch size (see Figure A2, between (e) and (f)) or it stays the same while sharpness increases (see Figure A4), which contradicts your claim on page 5 "larger batches yield systematically higher plateau levels".

*(iii) Unsupported or not enough supported claims*

 Many statements are not really justified by the author's paper. In Section 5:

"Batch Sharpness, not $\lambda\_{\text{max}}$ , governs the transition" The soundness of this claim is not clear to me. Is Figure 5 supposed to corroborate this statement ? $\lambda_{max}$ seems to change as well.

"Stabilization becomes distributional." The claim seems a bit weird to me, "can depend on the distribution of mini-batch Hessians". What do you mean by can ? Do you mean that it depends on the distribution in some cases, or that there is a possibility that there is such dependence ? I do not see how it is an implication of your work.

"Limits of “GD + noise” and diffusion/SDE surrogates." The failure of continuous models to capture EoS dynamics has been discussed thoroughly in [1] in the deterministic setting, I think it is worth mentioning. And your work did not specifically investigated it, for instance by comparing things such as trajectories deviation or sharpness behavior.

"Training happens in the intermediate regime. Moreover, we show that most practical training pipelines lie in the intermediate regime" Your work does not explicitly justify this claim. I think the paper should be more explicit about how it is justifying this claim.

"Hyperparameters must be tuned jointly." I think it should be supported by experiments.

Some findings could have been illustrated or supported by experiments, by example the fact that in the small batch regime momentum enforces stricter control on the curvature, which can be illustrated with a comparison with vanilla SGD

*Mild remark*

I think the appendix, in particular the experiments section, should be more polished. In Appendix G, sometimes there are color switch between batch sharpness and lambda max. In Figure A3, many axis label are missing or inconsistent in their presentation. In this same figure, the number of training step is not always the same in an incoherent way: from (g) to (i) there are way less steps compared with others, such that batch sharpness is barely entering or still have not enter the EoS regime.


[1] Understanding Optimization in Deep Learning with Central Flows, Jeremy M. Cohen, Alex Damian, Ameet Talwalkar, J. Zico Kolter, Jason D. Lee

---

> ### Author Rebuttal · Authors · 2026-03-30
>
> We thank you for your feedback, which greatly helped improve our paper. We have revised the paper throughout to improve clarity and make vague phrasing in Sections 2–5 more explicit. We also revised the discussion of related work and the EoSS framework to make the exposition more self-contained, as we recognized it sometimes assumed too much familiarity with prior works. Below we discuss our changes specifically.
>
>
> **(i) Writing and clarity.** Representative changes include: 'one-sided instability certificate' is now explicitly stated as a sufficient condition for instability, with a pointer to A.2; dense shorthand like 'temporal filtering and noise-shaping' has been unpacked; terms such as 'stabilizing perturbations,' 'instability boundary,' and 'mid-training' are now defined on first use; and SGDM(η, β, b) and SGD(η/(1−β), b) are now properly defined. We have also moved less central material (e.g., 'systems/implementation interactions') to the appendix, per your suggestion.
>
>
> In sections 4-5, we’d like to clarify that “the explicit condition on $\bar{H}$" refers to $\bar{H}$ entering the stability condition directly, connecting the stability boundary to the empirically similar $\lambda_{\max}$ behavior of SGDM and SGD at matched $\eta_{\mathrm{eff}}$.
>
>
> **(ii) About the main claim.** Stabilization at the predicted bound is **approximate** (Fig A4, A5), consistent with prior work: Cohen et al. observed approximate stabilization in the original EoS setting, and Andreyev and Beneventano found that Batch Sharpness stabilizes slightly below $2/\eta$ for small batches on CNNs. Critically, the $2/\eta$ boundary is still never crossed; the network simply does not re-enter progressive sharpening after hitting it, which we believe happens because it has exhausted the dataset's complexity after reaching EoSS. In particular, if we retrain on the full CIFAR (with correspondingly higher complexity budget) we can see that the dynamics stay at the EoSS (https://imgur.com/a/8cgmstS).
>
>
> The A1(a) plot is not informative about the claim: that run had not yet converged (loss is only at ~0.8). After running longer, Batch Sharpness stabilized at the claimed level (rerun https://imgur.com/a/3tsjfYs). Additionally, we acknowledge several figure errors: A2 (e)–(g) had wrong panels (fixed, now monotonic), A4 was mistakenly run without momentum (fixed), and the "mild remarks" are all now corrected. All corrected figures are consistent with the claimed results, across architectures, datasets, and hyperparameters.
>
> On the description of our stability analysis as 'weaker than usual': our lower bound is derived using standard linear-stability arguments from dynamical systems, consistent with the approach in Wu et al. [1], or Ma and Ying [2]. We show in section 4.1 that the stability bound in the noise dominated regime depends multiplicatively on $(1-\beta)$ in the small batch regime, and that is attained.
>
>
>
> **(iii) Unsupported claims.** We address the specific claims you highlighted below, and have clarified other statements throughout the paper accordingly.
>
>
> * *"Batch sharpness, not lambda max, governs the transition"* is directly supported by Figure 5: A change of batch size *instantaneously* changes batch sharpness but not $\lambda_{\max}$, since $\lambda_{\max}$ depends on the network weights, which are unchanged at that moment. Since $\lambda_{\max}$ remains the same, the catapult triggered indicates that $\lambda_{\max}$ does not govern stability. We have revised this paragraph to make the argument more explicit.
>
>
> * *"Stabilization becomes distributional"*: Both Batch Sharpness and the second-moment operator depend on higher moments of the mini-batch Hessian/sharpness distribution, not only on its first moment. This is explained in the subsequent sentence but has been revised to make it clearer.
>
> * *"Limits of GD + noise..."*: was insufficiently discussed and we have removed the claim.
>
> * *"Training happens in the intermediate regime"* : This simply means that standard CIFAR-10 batch sizes (32–1024) fall in the intermediate range between our small- and large-batch regimes, which we now state explicitly with supporting references.
>
> * *“Hyperparameter tuning”*: Removed, as this is a practical prescription rather than a result of our analysis.
>
> * "*Why Batch Sharpness rather than batch-wise sharpness or its expectation?*" – Andreyev and Beneventano (Appendix J.2) show that the expectation of batch-wise sharpness does not govern SGD stability. Therefore, it cannot govern SGDM/SGDN stability either. We have made this more explicit in our paper.
>
>
> **(iv) Additional experiments (ViT model and NLP task).** See "eval" response to LvgH.
>
>
>
> We hope we fully addressed your concerns. If any further questions arise, or if there is anything else we can do to be confident in increasing your score, please let us know. Thank you again!
>
> [1] "How SGD Selects..." (Wu, Ma, et al.)
>
> [2] "On Linear Stability of SGD..." (Chao Ma and Lexing Ying)

---

> > ### Author Rebuttal · Reviewer_nDAU · 2026-04-01
> >
> > Thank you very much for your detailed answer. I note that the authors are very open to feedback, which is greatly valuable.
> > One of my concern was that writing was, to me, too unclear at many places of the paper. I pointed several examples to the authors, and I am fairly convinced they understood my point. I trust the authors that they have enhanced the educational value of the article. I thank the authors for the clarification about the (previously long) list of claims, and I appreciate your acknowledgement that some of them were not sufficiently substantiated. I am happy to raise my score.

---

### Official Review · Reviewer_ZNZV · 2026-03-08

**Soundness:** 3
**Presentation:** 3
**Significance:** 4
**Originality:** 3
**Overall Recommendation:** 5
**Confidence:** 4

**Summary:**

This paper empirically investigates the effect of momentum on the EOS dynamics. It finds that the critical learning rate is often different from that of pure SGD, and its value depends on the momentum parameter $\beta$ as well as the batch size. For small batch sizes, the  critical learning rate for momentum SGD matches the theoretical expectation. In addition, it observed a new critical learning rate that is not explained by the current theory in the large batch size settings.

**Compliance With Llm Reviewing Policy:**

Affirmed.

**Final Justification:**

After discussion, I would like to keep my score.

**Key Questions For Authors:**

Instead of the concept of batch sharpness, is it possible to present the findings in terms of critical (effective) learning rate? The concern I have is that the BS formulas (eq. 1 and 2) could be misleading: it seems to suggest momentum methods lead to flatter (or sharper, for eq 2) regions than pure SGD (if one doesn’t notice the difference in learning rate), and indicates better (or worse) generalization performance if one is thinking about the connection between sharpness and generalization. However, the learning rate of SGD and momentum methods are often different, for example, momentum methods using a smaller learning rate to compensate for the factor $1-\beta$, such as in figure 4.

Do you have any theoretical intuition on why BS is larger (factor $1+\beta$) instead of smaller (factor $1-\beta$) in the large batch regime?

**Limitations:**

yes

**Strengths And Weaknesses:**

Strengths:

The experimental findings of the new critical learning rate for momentum SGD are very interesting and highly related to the current focus of the deep learning community, especially on the training dynamics. These empirical findings add a different perspective on the EOS phenomenon, I believe it can be helpful towards a final full understanding of EOS.

The experiments are well designed, investigated for different algorithms and batch sizes, and their results show clear signals and are impressive.

The empirical observations in the small batch size regime match the theory that momentum SGD is essentially pure SGD with a larger effective learning rate $\eta/(1-\beta)$. The paper also theoretically confirmed this.

The different behavior in the large batch size regime remains mysterious, it might suggest a new theoretical direction for further understanding momentum SGD with large batch sizes.


Weaknesses:


In the contribution list in introduction, item #6 is a duplication of #4. In addition, item #1 and #2 are also roughly talking about the same thing.

---

> ### Author Rebuttal · Authors · 2026-03-31
>
> We thank the reviewer for the positive assessment and the interesting technical questions. We address each point below.
>
> ***(1) "In the contribution list in introduction, item #6 is a duplication of #4."***
> We agree that these points overlap and will merge the two contributions to avoid redundancy.
>
> ***(2) "In addition, item #1 and #2 are also roughly talking about the same thing."***
> We agree that these contributions are closely related and could potentially be merged if we are running low on space. However, we intentionally separated them to distinguish two aspects:
> #1 aims to establish the existence an EoSS-like regime overall for SGDM/N,
> #2 characterizes the concrete batch-size dependent stabilization levels.
> That said, the distinction may not be important to the reader, and we are open to merging them in the revision for conciseness.
>
> ***(3) "is it possible to present the findings in terms of critical (effective) learning rate?":***
> Hypothetically this would be possible by inverting the instability conditions. However, the goal of our work is primarily descriptive, to characterize training dynamics, rather than prescriptive (selecting learning rates, etc.). For this reason we chose to present the results in terms of boundaries and regimes. Also, since we don’t consider learning rate scheduling, the critical learning rate formulation wouldn’t substantially improve interpretability. That said, this perspective may be useful and we will note it in the revision.
>
> ***(4) “it seems to suggest momentum methods lead to flatter (or sharper, for eq 2) regions than pure SGD”:***
> Yes, for fixed hyperparameters; we tried to clarify this both mathematically and empirically. In practice, SGD and SGDM use different learning rates: "However, the learning rate of SGD and momentum methods are often different, for example, momentum methods using a smaller learning rate to compensate for the factor 1− β , such as in figure 4." Our work provides a formal explanation for this practice. We will make it more explicit in the revision that the different stability levels are precisely why practitioners use different learning rates for SGD and SGDM.
>
> ***(5) "Do you have any theoretical intuition on why BS is larger (factor 1 + β) instead of smaller (factor 1 − β) in the large-batch regime?"***
> In the large-batch limit, the stochastic contribution to the stability boundary vanishes, and SGDM/SGDN approach the classical full-batch regime. In this deterministic regime, momentum recovers its classical stability-enlarging role. For example, for heavy-ball momentum, the admissible curvature increases from 2/η for GD to 2(1+β)/η, which explains why the batch-sharpness plateau increases rather than decreases at large batch sizes. This deterministic picture is classical for quadratic momentum methods (Polyak/Nesterov; see also Goh, 2017), and prior work (Cohen et al) connect the same threshold picture to EOS-like behavior in full-batch deep-network training.
>
> We already reflect this intuition in the paper: we state that the large-batch regime recovers full-batch behavior, and the 1D warm-up (Appendix) derives the interpolation between the stochastic and deterministic limits, with the noise-dominated threshold reducing to 2(1 − β)/η and the large-batch limit recovering the classical deterministic threshold. Equivalently, momentum can be interpreted as filtering fast oscillatory components along the sharpest eigendirection, consistent with the deterministic analysis. If this connection is not explicit enough in the current draft, we will add a clarification in the appendix.
>
> ***Concluding:***
> We appreciate the feedback and will revise the paper to reduce redundancy in the contribution list, better highlight the theoretical intuition behind the two regimes, and incorporate the suggested clarifications.

---

> > ### Author Rebuttal · Reviewer_ZNZV · 2026-04-04
> >
> > Thanks for the clarification. My concerns are clear.

---

### Official Review · Reviewer_LvgH · 2026-03-12

**Soundness:** 3
**Presentation:** 3
**Significance:** 3
**Originality:** 4
**Overall Recommendation:** 5
**Confidence:** 4

**Summary:**

This paper focuses on explaining the inner theretical rules of the optimization algorithms based on the momentum SGD for neural network training problem. The key contribution of this paper is that the authors find the sharpness at the edge of stochastic stability is constrained by the SGD momentum. More specifically, batch sharpness stabilizes in two distinct regimes. Experiments verify the effectiveness of the findings.

**Compliance With Llm Reviewing Policy:**

Affirmed.

**Key Questions For Authors:**

1. Can the findings in Section 3.3 be described by a mathematical expression?
2. Could you please restate the conclusions, as there are so many key points?
3. A notation table would be helpful. Otherwise, general readers may always get confused, as there are so many variables without definition.

**Limitations:**

1. The verification examples are insufficient; the study could include more domains, such as computer vision and natural language processing.
2. The findings appear difficult to apply in practical settings, as they offer limited guidance for hyperparameter tuning.
3. As the authors note, the terms "small-batch" and "large-batch" lack clear definitions. A precise definition or threshold should be provided (the authors only offer experiential definitions, such as b <= 16).
4. The absence of a mathematical expression to describe the findings in Section 3.3 represents a limitation.
5. A brief discussion, or even preliminary speculation, on how the findings on momentum might extend to adaptive methods (e.g., Adam) would make the conclusion more forward-looking and impactful.

**Strengths And Weaknesses:**

Strengths:
1. The theoretical principles underlying the optimization algorithm are revealed and explored. The theoretical analysis is both meaningful and insightful.
2. A key finding is the identification of a regime reversal. In the small-batch regime, momentum tightens the effective stability constraint derived in the study. In the large-batch regime, momentum recovers its classical role of enabling training in regions of higher curvature.
3. The work concludes that momentum does not simply shift the edge of stability, but rather makes it batch-dependent. This finding has significant implications for understanding optimization dynamics and guiding hyperparameter tuning.

Weaknesses:
1. The verification examples are insufficient; the study could include more domains, such as computer vision and natural language processing.
2. The findings appear difficult to apply in practical settings, as they offer limited guidance for hyperparameter tuning.
3. As the authors note, the terms "small-batch" and "large-batch" lack clear definitions. A precise definition or threshold should be provided (the authors only offer experiential definitions, such as b <= 16).
4. The absence of a mathematical expression to describe the findings in Section 3.3 represents a limitation.
5. A brief discussion, or even preliminary speculation, on how the findings on momentum might extend to adaptive methods (e.g., Adam) would make the conclusion more forward-looking and impactful.

---

> ### Author Rebuttal · Authors · 2026-03-31
>
> We thank the reviewer for the careful reading and for highlighting the originality of the batch-dependent stability picture. The main points seem to be (i) evaluation breadth, (ii) practical guidance, (iii) the meaning of “small-batch” and “large-batch,” and (iv) the status of Sec. 3.3 relative to the theory.
>
> ***On evaluation breadth:*** we agree the external scope is limited, and the paper already states this in Sec. 6.1 (p. 8, lines 430–437). At the same time, the current submission is broader than a single toy setting: Sec. 3.1 already summarizes that the same qualitative picture is observed across architectures, activations, and hyperparameter sweeps (p. 4 line 211 to p. 5 line 244), and the appendix includes MLP/CNN/ResNet and ReLU/SiLU ablations (Apps. G–H, pp. 24–36). So this is a limitation of breadth, but it is local to domain coverage rather than a weakness of the internal empirical pattern established in the submission.
>
> To address this directly and per your request, we are adding new experiments that expand both architectural and domain coverage: a ViT on CIFAR-10 (https://imgur.com/a/Puipr5Z), which represents a fundamentally different architecture based on attention rather than convolution, and SST-2 sentiment classification with CNN (https://imgur.com/a/TgRqe4p) and MLP (https://imgur.com/a/X9dSYTS), verifying that the two-regime behavior extends beyond the vision domain into NLP.
>
> ***On practical guidance:*** we agree this is not yet a prescriptive tuning theory. The supported takeaway is narrower: the effective instability margin depends jointly on (η,β,b), so settings that are stable in one batch regime need not transfer unchanged to another. This is exactly what the intervention results in Sec. 3.3 show: changing η, β, or b changes whether training catapults or resumes progressive sharpening (p. 6, lines 275–320; App. E, pp. 20–22). We do not mean to claim a complete recipe for test performance or generalization, and we will revise the conclusion accordingly.
>
> ***On “small-batch” and “large-batch”:*** we agree these should be defined operationally, not by an absolute cutoff. The paper already gives the two limiting laws in Sec. 3.1: small-batch corresponds to a plateau near 2(1−β)/η, and large-batch to the deterministic/full-batch momentum threshold (p. 5, lines 220–244). The same distinction appears analytically in Eq. (10), whose deterministic and stochastic terms interpolate between the two limits (p. 7, lines 347–359). The remark “b≲16” was intended only for the 8k CIFAR-10 subset (p. 4, line 219), not as a universal threshold; in revision we will replace it with the operational regime definition.
>
> ***On Sec. 3.3:*** we do not intend to claim a closed-form law for the full transition regime. The current paper already says that for momentum, Batch Sharpness is an empirical indicator rather than a proved certificate (p. 5, lines 246–261), and that a complete instability theory remains open (p. 6, lines 321–328). The mathematical statements we do claim are the limiting plateau laws plus the mean-square stability result supporting the small-batch scaling (Theorem 4.1, p. 7, lines 368–380; proof in App. D, pp. 18–19). A concise way to formalize Sec. 3.3 is: if a mid-training change in (η,β,b) lowers the operative threshold below the current BS, we observe a catapult; if it raises the threshold above the current BS, progressive sharpening resumes. This is exactly the pattern shown in Sec. 3.3/App. E.
>
> We also agree that a notation table and a shorter conclusion would improve readability, and we will add both. To that end, the findings distill into two main contributions:
>
> 1. **Empirical: batch-size-dependent EoSS under momentum.** SGDM and SGDN self-organize near a Batch Sharpness plateau whose level depends on batch size, interpolating between 2(1−β)/η (small-batch) and the deterministic momentum threshold (large-batch). Intervention experiments confirm this is an operative stability boundary.
>
> 2. **Theoretical: small-batch effective-step-size reduction.** Theorem 4.1 shows that in the noise-dominated regime, SGDM with (η, β) has the same stability condition as vanilla SGD with step size η/(1−β), explaining the 2(1−β)/η law and the batch-driven transition.
>
> In revision we will reorganize the conclusion around this two-part structure. Finally, we will expand the future-work positioning on adaptive methods / Muon in Sec. 6.2 (p. 8, lines 438–439).
>
> We hope we addressed fully your points and your questions. If any further questions arise, or is there anything else we can do for you to be confident in increasing your score, please let us know. Thank you again!

---

> > ### Author Rebuttal · Reviewer_LvgH · 2026-04-01
> >
> > Thanks for the reply. My concerns are answered.

---

### Official Review · Reviewer_JvTv · 2026-03-13

**Soundness:** 3
**Presentation:** 3
**Significance:** 2
**Originality:** 2
**Overall Recommendation:** 4
**Confidence:** 3

**Summary:**

This paper studies sharpness at the edge of stochastic stability. The authors start from the observation that momentum terms are added to modern optimizers and help improve the training in many aspects, but the mechanism is not well understood. Then they start from understanding the edge of stability phenomenon widely observed in the community and provide some metrics like batch sharpness to evaluate the training instability. Numerical experiments are given to further support their claims.

**Compliance With Llm Reviewing Policy:**

Affirmed.

**Key Questions For Authors:**

1. Could the authors provide some brief summary of the contributions? The current contributions contain many bullet points and may need some summary to highlight the main findings.

2. Theorem 4.1 only gives the results for mean-square stability. Could the authors provide the formal definition somewhere before the theorem? On the other hand, the theoretical results may not explain all the empirical observations in this paper.

3. Muon optimizer as mentioned by the author in Section 6.2 may need some citations and discussions.

**Limitations:**

1. The current theory in the paper is Theorem 4.1, which only gives the condition for mean-square stability. It may not explain other training instability such as edge of stochastic stability from the theoretical perspective.

2. Experiments are limited to some small models like MLP and ResNet.

**Strengths And Weaknesses:**

Strengths:

1. This paper provides some useful metrics such as batch sharpness to measure the training instability of SGDM and SGDN. They provide comprehensive empirical results to showcase the edge of stochastic stability phenomenon, as well as two regimes: large-batch and small-batch regimens that the momentum based methods operate in.


Weaknesses:

1. As also mentioned by the authors, the experiments are limited to some small models on CIFAR-10 and SVHN.

---

> ### Author Rebuttal · Authors · 2026-03-31
>
> We thank the reviewer for the positive assessment and the constructive suggestions. The main points seem to be (i) the contribution summary, (ii) the scope of Theorem 4.1 relative to the empirical findings, and (iii) the experimental breadth.
>
> On the contribution summary: the paper’s message is two-part. Empirically, we show that SGDM and SGDN exhibit EoSS-like dynamics with batch-dependent Batch-Sharpness plateaus (Intro/Contributions, p. 2, lines 73–107; Sec. 3.1, p. 4 line 211 to p. 5 line 244). Theoretically, we explain the small-batch law through mean-square stability: Eq. (10) gives the 1D interpolation between deterministic and stochastic terms (p. 7, lines 347–359), and Theorem 4.1 shows that in the noise-dominated regime SGDM(η,β) reduces to SGD with effective step size η/(1−β) (p. 7, lines 368–380; proof in App. D, pp. 18–19, lines 935–1002). We agree that the current bullet list is too granular, and in revision we will replace it with this shorter two-part summary.
>
> On Theorem 4.1: we agree its role should be stated more explicitly. The current paper already distinguishes the empirical and theoretical claims: Sec. 3.1 states that for momentum, Batch Sharpness is “an empirical indicator,” not a certificate (p. 5, lines 246–261), and Sec. 3.3 explicitly says that a complete divergence theorem for momentum remains open (p. 6, lines 321–328). Thus, the intended claim is narrower than “Theorem 4.1 explains all observed phenomena”: the theorem supports the small-batch scaling, while the broader EoSS claim is established empirically through the intervention experiments in Sec. 3.3 and App. E (p. 6, lines 275–320; pp. 20–22, lines 1045–1154). We also agree that mean-square stability should be defined immediately before Theorem 4.1, and we will add that formal definition.
>
> On Muon and scope: Sec. 6.1 already flags the evaluation limitation to CIFAR-10/SVHN and relatively small models (p. 8, lines 430–437), while the current submission still covers SGDM and SGDN, multiple architectures/activations, and broad (η,β,b) sweeps (Sec. 3.1, p. 4 line 211 to p. 5 line 244; Apps. G–H, pp. 24–36). We will move this scope statement earlier so it is read as a limitation of breadth rather than of soundness. Our goal was to extend this line of work from vanilla SGD to momentum methods under comparable conditions. That said, validating on larger-scale models is an important next step. To address this directly, we are adding a ViT on CIFAR-10 (https://imgur.com/a/Puipr5Z) and SST-2 sentiment classification with CNN (https://imgur.com/a/TgRqe4p) and MLP (https://imgur.com/a/X9dSYTS), extending coverage to attention-based architectures and NLP. We also agree that the brief Muon mention in Sec. 6.2 (p. 8, lines 438–439) should be better cited and discussed.
>
> We hope we addressed fully your points and your questions. If any further questions remain, or is there anything else we can do for you to be confident in increasing your score, please let us know. Thank you again!

---

> > ### Author Rebuttal · Reviewer_JvTv · 2026-04-05
> >
> > Thanks for the rebuttal, I tend to maintain the score for weak accept at this stage, and will discuss with other reviewers and AC later.

---

### Decision · Program_Chairs · 2026-04-30

**Decision:**

Accept (regular)

**Comment:**

**Recommendation: Accept**

This paper extends the Edge of Stability analysis to mini-batch SGD with Polyak and Nesterov momentum, using Batch Sharpness as the governing diagnostic to identify two batch-size-dependent plateau regimes: a noise-dominated small-batch regime in which Batch Sharpness stabilizes at $2(1-\beta)/\eta$ (strictly flatter than vanilla SGD) and a deterministic large-batch regime that recovers the classical $2(1+\beta)/\eta$ momentum threshold.

A one-dimensional mean-square stability analysis supports the small-batch law by reducing SGDM$(\eta,\beta)$ to SGD with effective step $\eta/(1-\beta)$, and mid-training intervention experiments on $\eta$, $\beta$, and $b$ certify the two plateau levels as operative instability boundaries rather than coincidental statistics. The four reviewers were uniformly positive (two Accept, two Weak Accept), highlighting the novelty of the regime-reversal finding (that momentum enforces *tighter* curvature control in the noise-dominated regime) together with the care taken in the intervention experiments and the cleanness of the small-batch theoretical result. I agree with this consensus and recommend acceptance.

Two issues from the discussion should be carried through to the camera-ready. First, reviewer nDAU raised substantive clarity concerns about Sections 2--5, which rely on dense or undefined terminology:
* ``one-sided instability certificate,''
* ``stabilizing perturbations,''
* ``instability boundary,''
* ``operator-centric,''
* ``operative plateau level,''
* and the SGDM$(\eta,\beta,b)$ notation itself

This is hard to parse without deep familiarity with prior EoS work. The authors committed to unpacking this vocabulary, adding a notation table, and compressing the contribution list into a two-part empirical/theoretical summary; please ensure these rewrites land in the final version. Second, the appendix figure corrections identified in the rebuttal --- the A2(e--g) panels that were non-monotonic, the A4 figure mistakenly run without momentum, and the A1(a) run that had not yet converged to the predicted plateau --- should be replaced with the corrected or re-run versions the authors produced during the discussion, since the small-batch bound claim rests on them.